# Towards improving saliency map interpretability using feature map smoothing

## Abstract

Input-gradient-based feature attribution methods, such as Vanilla Gradient, Integrated Gradients, and SmoothGrad, are widely used to explain image classifiers by generating saliency maps. However, these methods struggle to provide explanations that are both visually clear and quantitatively robust. Key challenges include ensuring that explanations are sparse, stable, and faithfully reflect the model's decision-making. Adversarial training, known for enhancing model robustness, have been shown to produce sparser explanations with these methods; however, this sparsity often comes at the cost of stability. In this work, we investigate the trade-off between stability and sparsity in saliency maps and propose the use of a smoothing layer during adversarial training. Through extensive experiments and evaluation, we demonstrate this smoothing technique improves the stability and faithfulness of saliency maps without sacrificing sparsity. Furthermore, a qualitative user study reveals that human evaluators tend to distrust explanations that are overly noisy or excessively sparse—issues commonly associated with explanations in naturally and adversarially trained models, respectively and prefer explanations produced by our proposed approach. Our findings offer a promising direction for generating reliable explanations with robust models, striking a balance between clarity and usability.

## 1 Introduction

Input gradient-based explanation methods highlight the features most influential to a model's decision by calculating the gradient of the model's output with respect to its input, visualized as saliency maps in images. One of the earliest approaches, Vanilla Gradient (VG) (Simonyan et al., 2014), computes gradients across input pixels, ranking features by their gradient magnitude. While prior studies have shown that input-gradients can capture relevant information regarding a model output (Samek et al., 2016), VG suffers from noisy saliency map. Hence, various methods like Integrated Gradient (IG) (Sundararajan et al., 2017), and SmoothGrad (SG) (Smilkov et al., 2017) have been proposed that modifies the input-gradient approach to reduce saliency map noise and improve the visual quality of the explanations.

However, quality explanations require more than visual appeal. Explanations should be comprehensible to users and satisfy quantitative measures to ensure their practical utility. Key properties include sparsity, which ensures explanations focus on the most relevant features by discarding irrelevant ones (Chalasani et al., 2020); stability, which guarantees consistent explanations across small input perturbations (Alvarez-Melis & Jaakkola, 2018); and faithfulness, ensuring that the explanations accurately reflect the model's actual decision-making process (Rong et al., 2022). These attributes are essential for explanations to be trustworthy and actionable in real-world applications.

In this work, we demonstrate a way to enhance above-mentioned properties of explanations in input-gradient based methods. We consider three representative input-gradient based methods (Vanilla Gradient (VG), Integrated Gradient (IG), and SmoothGrad (SG)) and first demonstrate that the stability of their explanations is closely tied to the model's sensitivity to input perturbations. Adversarial training (Goodfellow et al., 2015), a technique commonly used to improve model robustness, results in explanations that are sparser, aligning with previous studies (Chalasani et al., 2020; Etmann et al., 2019). However, we observe that this increased sparsity comes at the cost of reduced stability in explanations. To mitigate this trade-off, we introduce a smoothing layer applied during

adversarial training. Our extensive experiments with FMNIST, CIFAR-10 and ImageNette demonstrate that including feature-map smoothing using local filters like mean, median or Gaussian during adversarial training preserves stability and faithfulness of explanations without sacrificing on sparsity, resulting in explanations that are both clearer and more reliable.

*In addition*, we conduct a qualitative study to assess the comprehensibility of these explanations in human subjects. We interview 65 graduate students specializing in computer vision to assess their understanding of different types of explanations, which varies in terms of sparsity and smoothness. We use the Hoffman satisfaction scale as our assessment tool (Hoffman et al., 2023). Our findings reveal that explanations of input-gradient based attribution methods in naturally trained models are perceived as noisy and untrustworthy, while highly sparse explanations in adversarially trained models are also problematic due to the loss of information for enhancing sparsity. Explanations generated by input-gradient attribution methods for feature-map smoothed models are rated as more comprehensible, striking a balance between sparsity and clarity.

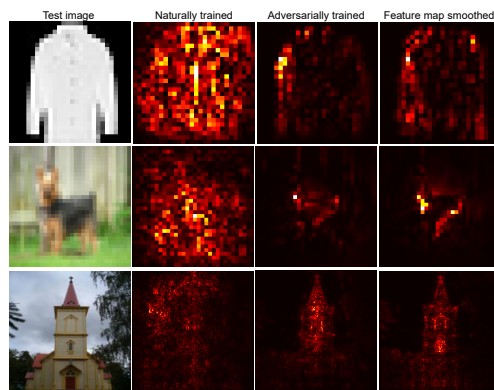

Figure 1: Saliency maps examples using Vanilla Gradient for different models that correctly classify the test images. Natural models produce noisy saliency maps ($2^{nd}$ column), adversarial models produce sparser maps ($3^{rd}$ column), and feature-map smoothed models smoothens the sparse maps ($4^{th}$ column), improving comprehensibility.

Figure 1 shows examples of saliency maps for different models on FMNIST, CIFAR-10, and ImageNette test images using Vanilla Gradient. We observe that saliency maps (a) for naturally trained models (second column) are noisy, and difficult to comprehend, (b) for adversarially trained models (third column) are sparse and align with the contours of the input image, but overly sparse saliency maps can lead to incomplete model understanding, and (c) for robust models with feature-map smoothing (fourth column) shows a reduction in sparsity to strike a balance between clarity and comprehensiveness. The smoothing helps reduce noise in the saliency map, resulting in explanations that are more continuous and coherent, while still maintaining a focus on key regions. Visualizations for Integrated Gradient, SmoothGrad and additional visualizations for Vanilla Gradient are provided in the Appendix L.

## 2 RELATED WORK

As highlighted by Ilyas et al. (2019), explanations that are meaningful and faithful to the model's decision-making process cannot be pursued independently from the training of the model, a principle central to our approach. Below we discuss such related works.

**Improving saliency maps by training modification**: Previous studies have proposed several modifications to model training to improve saliency maps. For instance, Kim et al. (2019) introduce layer-wise thresholding during backpropagation, while Dombrowski et al. (2019) suggest soft-plus activations as an alternative to ReLU for refining saliency maps. Wicker et al. (2023) develop a framework for certifying the robustness of explanations through training constraints. Meanwhile, Chenyang & Chan (2023) propose training object detectors by ensuring explanation consistency within same object and distinctions between different objects. In contrast, we do not make such modifications, and enhance the quality of explanations by applying simple smoothing filters during adversarial training.

**Study of saliency maps in robust models**: Some previous works have also explored saliency map quality in robust models (Etmann et al., 2019; Zhang & Zhu, 2019; Chalasani et al., 2020; Mangla et al., 2020; Shah et al., 2021), typically evaluating sparsity, or visual quality. Chalasani et al. (2020) show that adversarial training with $L_\infty$ attacks leads to sparse saliency maps, and theoretically demonstrate that training a 1-layer network by encouraging stability of explanations is equivalent to adversarial training, but do not present results on multi-layer networks. Etmann et al. (2019) ex-

plain the interpretability of robust models by demonstrating alignment between image and saliency maps, which works well for smaller datasets like MNIST but does not scale to larger datasets like ImageNet. Zhang & Zhu (2019) argue that adversarially trained models produce shape-biased representations, resulting in sparser saliency maps. In contrast, we approach the quality of saliency maps via the stability of the input-gradient explanation methods and establish a theoretical connection with model sensitivity, and propose adversarial training with feature-map smoothing as the mitigation of sparsity-stability tradeoff.

## 3 METHOD

**Preliminaries:** Consider a single-layer DNN with the form $F(\mathbf{x}) = H(\langle \mathbf{w}, \mathbf{x} \rangle)$, where $H$ is a differentiable scalar-valued activation function (e.g., sigmoid), $\langle \mathbf{w}, \mathbf{x} \rangle$ is the dot product between the weight vector $\mathbf{w}$ and input $\mathbf{x} \in \mathbb{R}^d$. The Vanilla Gradient (VG) method (Simonyan et al., 2014) measures the sensitivity of the model output $F(\mathbf{x})$ with respect to each feature of the input $\mathbf{x}$. This is given by computing the gradient of the output $F(\mathbf{x})$ with respect to the input $\mathbf{x}$. The Integrated Gradients (IG) method (Sundararajan et al., 2017) averages the gradients along a straight-line path from a baseline input $\mathbf{x}'$ (often a zero vector) to the actual input $\mathbf{x}$. SmoothGrad (SG) (Smilkov et al., 2017) improves on any gradient-based explanations like VG or IG by adding random noise to the input $\mathbf{x}$ multiple times, calculating the explanations for each noisy version, and then averaging the results.

### 3.1 RELATIONSHIP BETWEEN EXPLANATION STABILITY AND MODEL SENSITIVITY

In this section, we establish the foundation for understanding how model sensitivity affects the stability of gradient-based saliency maps. We first compute the explanation using VG, given by:

$$VG(\mathbf{x}) = \frac{\partial F(\mathbf{x})}{\partial \mathbf{x}} = \frac{\partial H(\langle \mathbf{w}, \mathbf{x} \rangle)}{\partial \mathbf{x}} = H'(\langle \mathbf{w}, \mathbf{x} \rangle).\mathbf{w} \tag{1}$$

Here, $H'(\langle \mathbf{w}, \mathbf{x} \rangle)$ is the gradient of activation function $H$ with respect to the $\langle \mathbf{w}, \mathbf{x} \rangle$. For eg, for a sigmoid activation function, $H'(z) = H(z)(1 - H(z))$ where $z = \langle \mathbf{w}, \mathbf{x} \rangle$. The feature attribution score computed by IG for feature $i$ of input image $\mathbf{x} \in R^d$ with baseline $\mathbf{u}$, model $F$ is given by Eqn. 2, with a closed form expression of Eqn. 3 (Chalasani et al., 2020):

$$IG_i^F(\mathbf{x}, \mathbf{u}) = (x_i - u_i). \int_{\alpha=0}^{1} \partial_i F(\mathbf{u} + \alpha(\mathbf{x} - \mathbf{u})) \partial \alpha \tag{2}$$

$$IG^F(\mathbf{x}, \mathbf{u}) = [F(\mathbf{x}) - F(\mathbf{u})] \frac{(\mathbf{x} - \mathbf{u}) \odot \mathbf{w}}{\langle \mathbf{x} - \mathbf{u}, \mathbf{w} \rangle} \tag{3}$$

For SG, we add Gaussian noise $\mathbf{n} \sim \mathcal{N}(0, \sigma^2)$ to the input $\mathbf{x}$ and compute the input-gradient for multiple noisy samples $\mathbf{x}_k = \mathbf{x} + \mathbf{n}_k$ for $k = 1, \ldots, N$, where $N$ is the number of noise samples. SG explanation, when aggregating VG, is given by:

$$SG(\mathbf{x}) = \frac{1}{N} \sum_{k=1}^{N} \frac{\partial F(\mathbf{x}_k)}{\partial \mathbf{x}_k} = \frac{1}{N} \sum_{k=1}^{N} \frac{\partial H(\langle \mathbf{w}, \mathbf{x}_k \rangle)}{\partial \mathbf{x}_k} = \frac{1}{N} \sum_{k=1}^{N} H'(\langle \mathbf{w}, \mathbf{x}_k \rangle).\mathbf{w} \tag{4}$$

Now consider $\mathbf{x}' \in \mathcal{N}_{\mathbf{x}}$ is a noisy version of input image $\mathbf{x}$ where $\mathcal{N}_{\mathbf{x}}$ indicates a neighborhood of inputs $\mathbf{x}$ where the model prediction is locally consistent. The stability of explanations-VG, IG and SG-can be computed by measuring the norm of the difference between the original explanation and explanation for the noisy image. Using Eqns. 1, 3 and 4, we obtain,

$$\Delta_{VG} = ||VG^F(\mathbf{x}') - VG^F(\mathbf{x})||_1 \leq (F(\mathbf{x}') - F(\mathbf{x})).\mathbf{w} \tag{5}$$

$$\Delta_{IG} = ||IG^F(\mathbf{x}', \mathbf{u}) - IG^F(\mathbf{x}, \mathbf{u})||_1 \approx ||IG^F(\mathbf{x}', \mathbf{x})||_1 = \left\|[F(\mathbf{x}') - F(\mathbf{x})] \frac{(\mathbf{x}' - \mathbf{x}) \odot \mathbf{w}}{\langle \mathbf{x}' - \mathbf{x}, \mathbf{w} \rangle}\right\|_1 \tag{6}$$

$$\Delta_{SG} = \sum_{k=1}^{N} ||SG^F(\mathbf{x}') - SG^F(\mathbf{x})||_1 \le \frac{1}{N}(F(\mathbf{x}') - F(\mathbf{x}))).\mathbf{w} \tag{7}$$

Since $\mathbf{w}$ is fixed for a given model, the bounds in Eqns 5, 6 and 7 indicate that the stability of explanations is influenced by the model sensitivity, setting up a basis for using methods that enhance explanation stability by reducing model sensitivity. However, these bounds do not serve a strict proportional relationship between model sensitivity and attribution stability, and should not be interpreted as such. Rather, the bounds serve as approximate indicators, highlighting that attribution stability is influenced by model sensitivity. See Appendix J for the complete proof and Appendix G for conditions affecting the tightness of the stability bounds.

## 3.2 ADVERSARIAL TRAINING AND IMPACT ON SALIENCY MAP STABILITY

Building on the observation from Section 3.1, we apply adversarial training (Goodfellow et al., 2015) as a method to address model sensitivity. Adversarial training modifies the loss to minimize the sensitivity to input perturbations, by solving $\mathbb{E}_{(\mathbf{x},y)\sim D}\left[\max_{||\delta||_\infty \le \epsilon} \mathbb{L}(\mathbf{x} + \delta, y; \mathbf{w})\right]$ where $\delta$ is a small perturbation and $\epsilon$ is the perturbation bound.

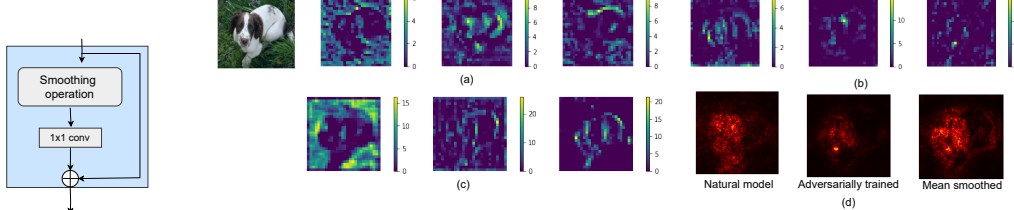

Figure 2: A feature-map smoothing block

Figure 3: Plot of feature maps (channel=7, 21, 127) after first residual block for a test image on different ResNet18 ImageNette models: (a) a naturally trained model, (b) an adversarially-trained model, (c) an adversarially-trained model with feature-map smoothing (mean filter) (d) corresponding saliency maps using Vanilla Gradient.

In Figure 3, given a test image from the ImageNette dataset, we visualize feature maps derived from (a) a naturally trained model and (b) an adversarially trained model. All models are trained using the identical ResNet18 architecture (He et al., 2016) and training settings. The visualized feature maps are taken after the first residual block, which has 128 channels, with maps from three representative channels shown for comparison. The notable difference between Figure 3(a) and Figure 3(b) is that many feature activations in the adversarially-trained model are shrunk, leading to more selective attention, which influences the saliency maps produced by input-gradient based methods. Such methods yield sparser explanations in the adversarially trained model compared to the natural model (see Figure 3(d)). This effect is also explored in (Etmann et al., 2019; Chalasani et al., 2020). However, intriguingly, adversarial training does not lead to improvement in explanation stability in DNNs. For such models, we find (in Sections 4.1 and 4.3) that while we gain sparsity in saliency maps, the sparser explanations affect explanation stability and comprehensibility.

## 3.3 FEATURE MAP SMOOTHING FOR COMPREHENSIBLE EXPLANATIONS

To address the limitations of adversarial training on saliency map stability and comprehensibility, we incorporate feature map smoothing (Xie et al., 2019). By smoothing out the sharp reductions in feature activations, these smoothing techniques help stabilize input-gradient-based explanations, producing saliency maps that are both sparse and stable, when combined with adversarial training.

In our study, we explore three local-smoothing filters (mean, median, and Gaussian) and two non-local smoothing filters (non-local Gaussian, and embedded Gaussian) (Wang et al., 2018) due to their complementary properties in smoothing feature maps (See Appendix B for details on each filter). Figure 2 represents a feature-map smoothing block, which can take any feature map as input. The block applies a smoothing operation, followed by a 1x1 convolutional layer, and combines the

Table 1: Sparsity-Stability Evaluation on Vanilla Gradient (VG), Integrated Gradient (IG) and SmoothGrad (SG). ↑ and ↓ indicate that larger & smaller values are better respectively.

| | | FMNIST | | | | | | CIFAR-10 | | | | | | ImageNette | | | | | |
|---|---|---|---|---|---|---|---|---|---|---|---|---|---|---|---|---|---|---|---|
| | | A | M1 | M2 | G | E | NG | A | M1 | M2 | G | E | NG | A | M1 | M2 | G | E | NG |
| VG | dG↑ | 0.198 | 0.198 | 0.171 | 0.183 | 0.188 | **0.219** | 0.188 | 0.185 | 0.181 | 0.185 | 0.189 | **0.190** | 0.050 | 0.018 | 0.036 | 0.063 | **0.117** | 0.107 |
| | dRIS↓ | 2.193 | 1.396 | **-1.025** | 1.168 | -0.400 | 1.781 | -0.458 | -0.621 | **-0.676** | -0.465 | -0.503 | -0.637 | -0.056 | **-0.121** | -0.016 | -0.098 | 0.767 | 0.401 |
| | dROS↓ | 2.084 | 1.121 | **-1.222** | 0.739 | -0.451 | 1.785 | 0.217 | 0.260 | **0.214** | 0.226 | 0.280 | 0.257 | -0.362 | **-0.470** | -0.297 | -0.456 | 0.386 | 0.240 |
| | dRRS↓ | 2.489 | 1.600 | **-0.799** | 1.452 | -0.126 | 2.202 | 0.445 | 0.453 | **0.433** | 0.457 | 0.438 | 0.467 | 0.241 | **-0.218** | -0.096 | -0.078 | 0.778 | 0.441 |
| IG | dG↑ | 0.067 | 0.075 | 0.047 | 0.050 | 0.021 | **0.069** | 0.091 | 0.091 | 0.092 | 0.094 | 0.087 | **0.095** | 0.034 | 0.033 | 0.062 | 0.041 | **0.063** | 0.056 |
| | dRIS↓ | 2.016 | 2.679 | **-0.843** | 4.564 | 1.007 | 2.714 | -1.056 | -1.504 | **-1.862** | -1.662 | -1.499 | -1.597 | 0.143 | **-0.0071** | 0.135 | 0.276 | 0.370 | 0.163 |
| | dROS↓ | 1.931 | 2.917 | **-0.698** | 4.681 | 2.103 | 2.526 | 0.228 | 0.350 | **-0.123** | -0.090 | 0.593 | 0.041 | -0.230 | **-0.532** | -0.451 | -0.376 | -0.273 | -0.038 |
| | dRRS↓ | 2.037 | 2.811 | **-0.741** | 5.030 | 1.622 | 2.676 | 1.050 | 0.219 | **0.163** | 0.410 | 0.258 | 0.243 | -0.157 | -0.121 | -0.027 | **-0.224** | 0.135 | -0.232 |
| SG | dG↑ | 0.198 | 0.198 | 0.171 | 0.183 | 0.158 | **0.219** | 0.681 | 0.684 | 0.684 | 0.678 | 0.684 | **0.686** | 0.036 | 0.028 | 0.064 | 0.035 | **0.101** | 0.068 |
| | dRIS↓ | 0.945 | 0.799 | **-0.466** | 0.994 | -0.282 | 2.015 | -0.040 | -0.034 | **-0.191** | 0.885 | 0.372 | 0.340 | 0.017 | **-0.148** | 0.719 | 0.045 | 0.272 | 0.030 |
| | dROS↓ | 5.593 | 3.418 | **-0.194** | 2.034 | 1.099 | 2.988 | 4.619 | 5.087 | **4.393** | 4.540 | 4.733 | 0.494 | -0.576 | **-0.728** | -0.589 | -0.657 | -0.331 | -0.348 |
| | dRRS↓ | -1.360 | 0.028 | -0.850 | -0.694 | **-2.085** | 1.245 | -2.561 | -2.469 | **-2.693** | -2.612 | -2.440 | -2.582 | -0.274 | **-0.381** | -0.216 | -0.306 | 0.010 | -0.234 |

result with the input through a residual connection. The introduction of this smoothing block has minimal effect on model accuracy (detailed in Appendix D), but it significantly alters the behavior of input-gradient based explanations.

As shown in Figure 3(c), the feature maps of an adversarially trained model with feature-map smoothing exhibit a noticeable smoothing effect, which varies based on the type of filter applied. For instance, with mean filtering, rapid changes in feature map values are reduced through averaging. While adversarial training alone (Figure 3(b)) suppresses many feature activations, the addition of smoothing helps preserve key features while eliminating the sharp discontinuities typically seen in naturally trained models. This results in smoother and more interpretable saliency maps, as demonstrated in Figure 3(d). The smoothed feature maps also align with the stability bounds derived in Section 3.1, as smoother activations reduce the norm $\|F(\mathbf{x}') - F(\mathbf{x})\|$, yielding tighter bounds for VG, IG, and SG. In Appendix H, we also discuss the effect of convolution operation on the receptive field expansion in the smoothing block and demonstrate that smoothing filters still provides a competitive advantage especially on sparsity and stability of saliency maps.

## 4 EXPERIMENT AND ANALYSIS

### 4.1 EXPERIMENT FRAMEWORK

**Setup:** We evaluate our approach on three datasets: FMNIST (Xiao et al., 2017), CIFAR-10 (Krizhevsky et al., 2009), and ImageNette (Howard, 2020), training several model variants for each. The variants include: 1) naturally trained (N), 2) adversarially trained (A), 3) adversarial training with mean-filter smoothing (M1), 4) adversarial training with median-filter smoothing (M2), 5) adversarial training with Gaussian-filter smoothing (G), 6) adversarial training with embedded filter smoothing (E), and 7) adversarial training with non-local Gaussian smoothing (NG). Following the setup from Chalasani et al. (2020), we use LeNet (LeCun et al., 1998) for FMNIST and Wide-ResNet (Zagoruyko & Komodakis, 2016) for CIFAR-10. We use ResNet-18 He et al. (2016)for ImageNette. For adversarial training, we apply perturbations under the $L_\infty$ norm using the PGD attack (Madry et al., 2018). The models are trained with $\epsilon = 0.1$ for FMNIST and CIFAR-10, and $\epsilon = 1/255$ for ImageNette, as these values yielded the best performance across our evaluations. We also achieved optimal results by adding the smoothing block after the first convolutional or residual block. Full details of our datasets and training methodology are provided in Appendix A, and we discuss the impact of altering the smoothing filter's position in Appendix E. We also discuss the effect of robust training strategy on saliency map quality for a different network architecture in Appendix F.

**Evaluation Metrics:** Given a saliency map from Vanilla Gradient (VG), Integrated Gradient (IG) and SmoothGrad (SG) for each model and dataset, we compute its sparseness using Gini index (G) (Chalasani et al., 2020), and its stability using relative input stability (RIS), relative output stability (ROS) and relative representation stability (RRS) (Agarwal et al., 2022). We analyze faithfulness using ROAD analysis (Rong et al., 2022), and saliency map similarity using structural similarity index (SSIM) (Adebayo et al., 2018). All results are aggregated for 1000 randomly selected test images that the model accurately classifies across all datasets. See Appendix K for detail discussion on metrics. Our code is available at https://anonymous.4open.science/r/iclr2025xai/README.md.

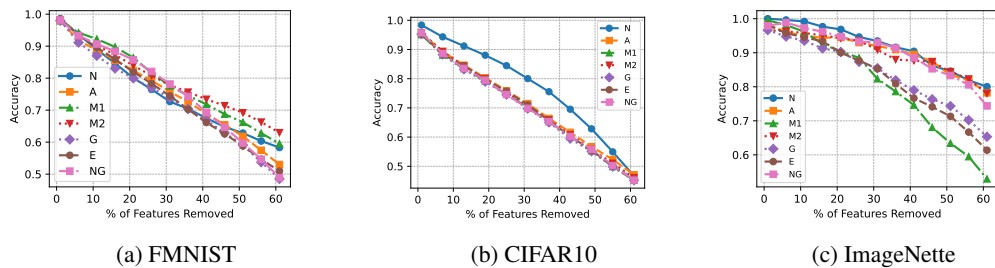

(a) FMNIST        (b) CIFAR10        (c) ImageNette

Figure 4: ROAD evaluation of VG with MoRF removal strategy

## 4.2 RESULTS AND DISCUSSION

**1. On the sparsity and stability of saliency maps:** Similar to Chalasani et al. (2020), we compare the sparsity and stability improvement of saliency maps with respect to the naturally trained model (N). Specifically, for a given training method (M), we compute the following metrics that quantify the improvement in sparseness (dG), relative input stability (dRIS), relative output stability (dROS), and relative representation stability (dRRS) of the explanation method $\phi(.) \in \{VG, IG, SG\}$: $dG[\phi(\mathbf{x})] = G^M[\phi(\mathbf{x})] - G^N[\phi(\mathbf{x})]$, $dRIS[\phi(\mathbf{x})] = RIS^M[\phi(\mathbf{x})] - RIS^N[\phi(\mathbf{x})]$, $dROS[\phi(\mathbf{x})] = ROS^M[\phi(\mathbf{x})] - ROS^N[\phi(\mathbf{x})]$ and $dRRS[\phi(\mathbf{x})] = RRS^M[\phi(\mathbf{x})] - RRS^N[\phi(\mathbf{x})]$.

As illustrated in Table 1, across all datasets-FMNIST, CIFAR-10 and ImageNette-robust models consistently achieve higher positive dG values for all three explanation methods (VG, IG, SG), indicating that these methods produce sparser saliency maps in robust models than naturally trained models. Notably, the highest sparsity gains in explanations are observed in models utilizing non-local smoothing filters. On FMNIST and CIFAR-10, the NG models (non-local gaussian) attain the highest sparsity across all explanation methods, and on ImageNette dataset, model E (embedded gaussian) achieves the highest sparsity for VG, IG and SG. However, this increase in sparsity comes at the expense of stability, as most robust models exhibit reduced stability in their explanations, suggesting an inverse relationship between the sparsity and stability of saliency maps. For example: NG models, while achieving high sparsity for explanations, show significant drops in dRIS, dROS, and dRRS, indicating that their explanations may be more sensitive to input perturbations or variations in model representations. Notably, models M1 and M2 provide a promising middle-ground. On FMNIST and CIFAR-10, explanations consistently achieves the highest stability in M2 across all methods, while still maintaining sparsity gain. On ImageNette, M1 offers the best stability across explanation methods. These results suggest that the use of local smoothing filters like mean and median filters during adversarial training can preserve the stability of saliency maps while maintaining a degree of sparsity.

**2. On the faithfulness of explanation:** Faithfulness metrics that involve pixel removal and measuring model prediction changes (such as insertion/deletion (Petsiuk et al., 2018)) introduces artifacts and cause a distribution shift in the perturbed inputs. Retraining based approaches like ROAR (Hooker et al., 2019) addresses this problem but is computationally expensive. ROAD (Rong et al., 2022) addresses both concerns in faithfulness evaluation by measuring model accuracy on the test set as pixels are iteratively removed using a nosily linear imputation strategy. We adopt the MoRF (Most Relevant First) removal strategy but ROAD demonstrates consistent results with both MoRF and LeRF (Least Removal First) removal strategy. For further details, see Rong et al. (2022).

Figure 4, 5 and 6 illustrate the evaluation results for VG, IG and SG using ROAD. In the MoRF strategy, a faster drop in accuracy with increase in removal of $k$ most important features indicate that key discriminative features are being removed. Across VG, IG and SG, on FMNIST (See Figure 4a, 5a, 6a), while natural models start with sharper drop in accuracy, robust models quickly surpass them. On CIFAR-10 and ImageNette, robust models exhibit sharper accuracy drops across VG, IG, and SG, suggesting these explanation methods capture more discriminative features from the input images. Furthermore, the application of smoothing filters enhances explanations differently across datasets. For eg. on ImageNette, the complex feature patterns cause smoothing filters to diverge in impact. This contrast is less prominent on CIFAR-10, where the simpler feature structures lead to more similar accuracy trajectories. This shows that even with the same explanation method, the faithfulness of the explanation is influenced by the model and dataset it is applied to.

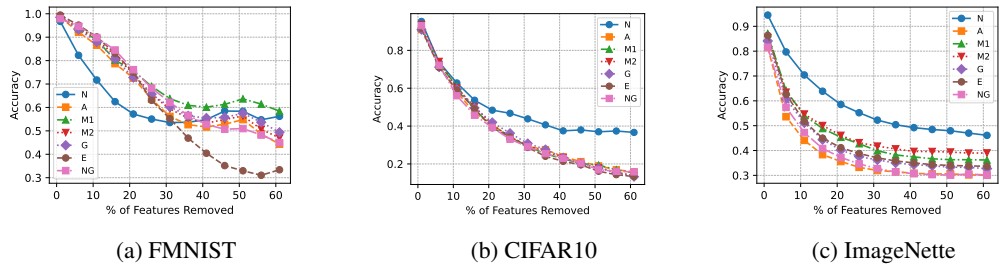

Figure 5: ROAD evaluation of IG with MoRF removal strategy

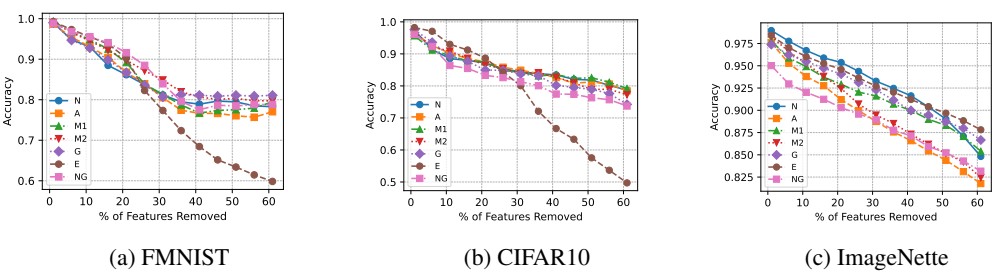

Figure 6: ROAD evaluation of SG with MoRF removal strategy

Additionally, we evaluate faithfulness using faithfulness estimate (Alvarez Melis & Jaakkola, 2018), Softmax Information Curve (SIC) (Kapishnikov et al., 2019), and Accuracy Information Curve (AIC) (Kapishnikov et al., 2019) in Appendix C. The results confirm that saliency maps from robust models are consistently more faithful compared to those from naturally trained models, and the introduction of smoothing filters does not affect explanation faithfulness.

**4. On the structural similarity of saliency maps:** Following Adebayo et al. (2018), we plot the structural similarity of attribution maps. For each image $\mathbf{x}$, we introduce Gaussian noise ($\mathcal{N}(0, \sigma)$) to create its noisy counterpart $\mathbf{x}'$ while ensuring consistent model predictions. Subsequently, we compute saliency maps for $\mathbf{x}$ and $\mathbf{x}'$ and measure the structural similarity between the maps. As illustrated in Figures 7, 8 and 9, the input-gradients of robust models exhibit greater invariance to noise compared to naturally trained models. This outcome aligns with expectations, as adversarially trained models undergo training with additional perturbation of input. The inclusion of feature map smoothing imparts an additional layer of invariance to noise, and can further improve the structural similarity of saliency maps. In SG (Figure 9), the saliency maps have similar structural similarity over different standard deviation of the noise distribution. This is because SG aggregates explanations by introducing noise to the given test image, so explanations are substantially more robust to input variations. However, saliency maps of robust models still outperform naturally trained models in structural similarity.

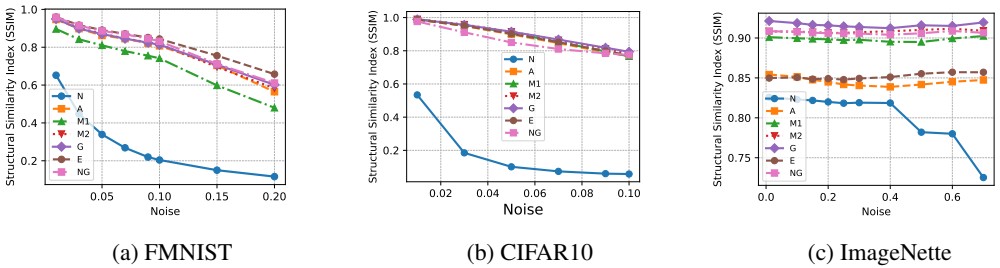

Figure 7: Structural similarity evaluation of VG

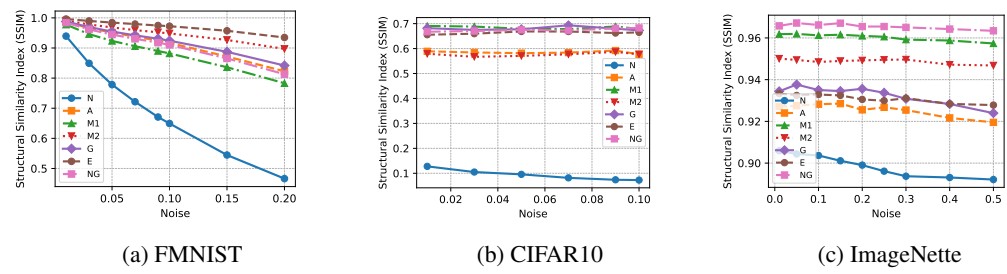

Figure 8: Structural similarity evaluation of IG

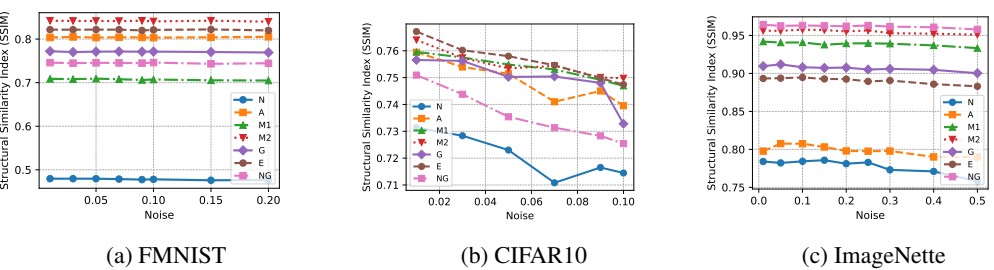

Figure 9: Structural similarity evaluation of SG

**5. Trade-off between model performance & saliency map quality:** Our findings reveal that: (a) input-gradient based attribution methods produce sparse saliency maps in adversarially trained models, (b) adversarially trained models with non-local-feature-map smoothing, increase the sparsity of saliency maps but compromise on stability, (c) adversarially trained models, with local-feature-map smoothing, enhances the stability of saliency maps without compromising on sparsity, (d) saliency maps in robust models demonstrate invariance to noise, and (e) saliency maps in robust models are more faithful to the underlying model than naturally trained counterparts. These observations lead to the conclusion that saliency maps in robust models are more reliable and interpretable than natural models for the input-gradient based attribution methods. However, it's important to note a caveat: such models come at the expense of benign accuracy.

We illustrate this tradeoff in Figure 10 and Figure 11. We train $L_\infty(\epsilon)$ robust models with perturbation strength $\epsilon \in [0.01, 0.03, 0.06, 0.1]$ for FMNIST and CIFAR-10 datasets. For each robust model, we compute its benign accuracy, and three saliency map characteristics using Vanilla Gradient: sparsity (Chalasani et al., 2020), faithfulness estimate (Alvarez Melis & Jaakkola, 2018), and structural similarity (Adebayo et al., 2018). Then, we plot the saliency map characteristics against the benign accuracy of the model. Figure 10 and Figure 11 illustrate that the higher the sparsity, faithfulness, and sensitivity, the lower the benign accuracy. This trend holds across all robust models, where increasing model robustness tends to reduce benign accuracy but enhances sparsity, faithfulness and

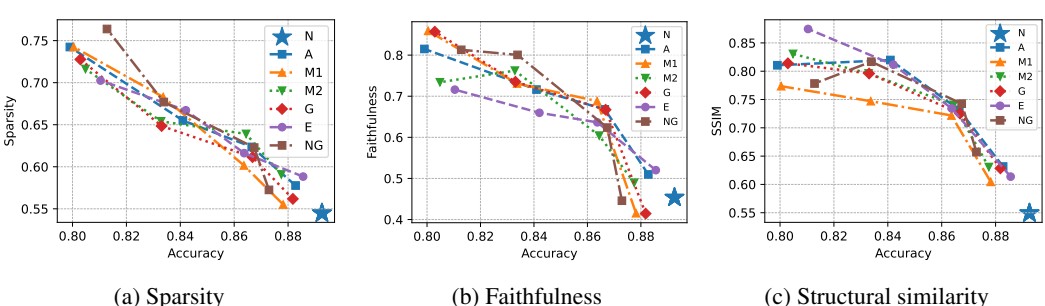

Figure 10: Tradeoff between saliency map quality and model performance on FMNIST

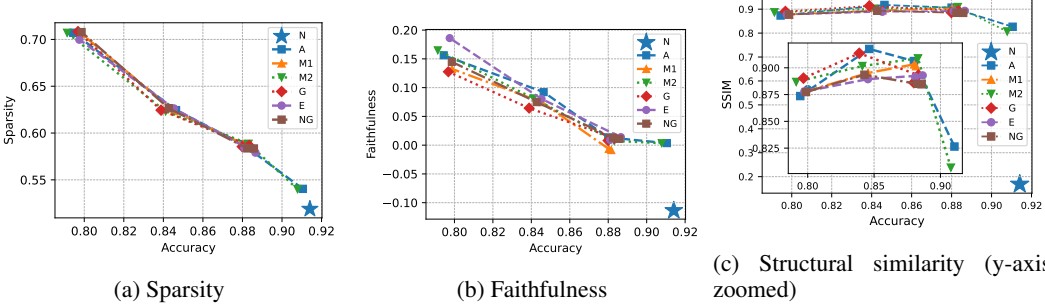

Figure 11: Tradeoff between saliency map quality and model performance on CIFAR-10

structural similarity of saliency maps. In contrast, naturally trained models have lower values of all three saliency map metrics but at much higher benign accuracy.

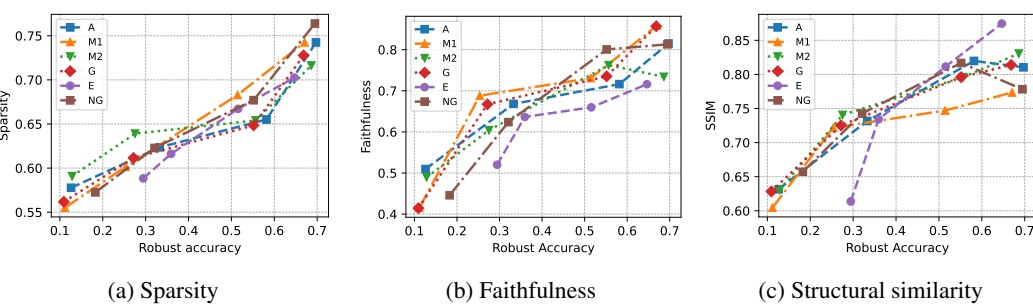

Figure 12: Relationship between model robustness and saliency map quality on FMNIST

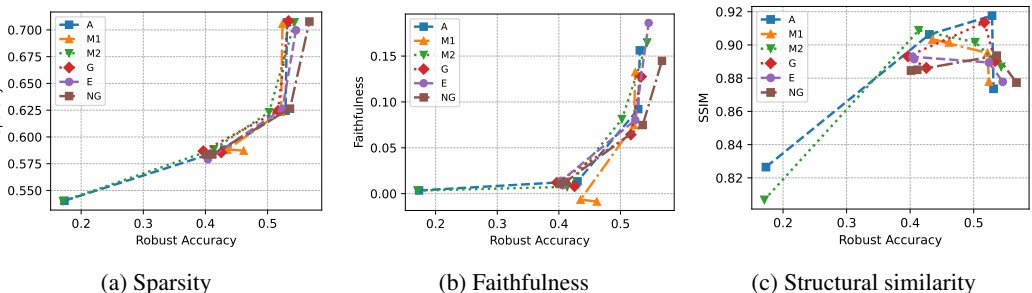

Figure 13: Relationship between model robustness and saliency map quality on CIFAR-10

**6. Relationship between model robustness & saliency map quality:** For each $L_\infty(\epsilon)$ robust model trained at $\epsilon \in [0.01, 0.03, 0.06, 0.1]$, we compute its robust accuracy as the accuracy of classifying PGD (Madry et al., 2018) samples, created at $\epsilon = 0.1$ and steps $= 100$. We plot the relationship between sparsity (Chalasani et al., 2020), faithfulness estimate (Alvarez Melis & Jaakkola, 2018), and structural similarity (Adebayo et al., 2018) against robust accuracy in Figure 12 and Figure 13, where we can observe that the sparsity, faithfulness, and sensitivity of saliency maps improves with the increase in the robustness of the model.

### 4.3 QUALITATIVE ANALYSIS

Our quantitative studies demonstrate that saliency maps in adversarially trained models are sparse but at the expense of stability. Incorporating local feature-map smoothing improves stability of saliency maps without drastically compromising sparsity, balancing these two aspects. In this section, we analyze how well end-users comprehend saliency maps from different model training strategies based on the level of sparsity.

We conducted an experiment with 65 graduate students (Ph.D./ Masters), each with at least a year of experience in computer vision[1]. The objective was to determine whether the information conveyed by saliency maps was sufficient for understanding and trusting the underlying model behavior. Participants were shown saliency maps using Vanilla Gradient from three models—naturally trained, adversarially trained, and adversarially trained with feature-map smoothing (median filter)—for 10 images across FMNIST and CIFAR-10 datasets, resulting in 60 image-saliency pairs. The saliency maps were presented in random order, and participants were unaware of the model that generated them. Afterward, they rated each saliency map using the Hoffman satisfaction scale (Hoffman et al., 2023), responding to two key questions: 1) "Does the explanation provide sufficient information?" and 2) "Do you trust the model's classification based on this saliency map?" Ratings were on a scale of 1 (strongly disagree) to 5 (strongly agree). Finally, participants were asked to compare saliency maps from all three models side by side and select the most comprehensible explanation, providing free-text justifications for their choices.

**Results:** We assessed the comprehensibility of the saliency maps based on two metrics: sufficiency and trust. For the naturally trained model, participants rated sufficiency at an average of 2.08 ($\pm$ 0.75) and trust at 2.02 ($\pm$ 0.82), indicating that the noisy maps from this model were generally considered untrustworthy. In contrast, adversarially trained models fared better, with sufficiency scoring 2.99 ($\pm$ 0.93) and trust 3.08 ($\pm$ 0.90), as participants found these maps clearer and more aligned with the images. The feature-map smoothed adversarial model scored the highest, with sufficiency at 3.33 ($\pm$ 1.03) and trust at 3.14 ($\pm$ 1.01). Participants appreciated the reduction in noise and highlighted the clarity and relevance of the explanations. When comparing saliency maps directly, 56% of participants preferred the maps from the feature-map smoothed model, 29% favored the adversarial model, and only 15% selected the naturally trained model. The majority cited reasons such as "highlighting important features without excessive detail" and "close enough to the image with the least noise".

To statistically validate the results, we performed Wilcoxon signed-rank test (Woolson, 2007) and one-way ANOVA (Cuevas et al., 2004) on the sufficiency and trust metrics across the three models. The p-values were extremely small ($< 0.001$), confirming significant differences between the models in terms of both metrics. This shows that the different training strategies lead to distinct levels of comprehensibility and trustworthiness in saliency maps. Details of qualitative study and results are provided in Appendix I.

## 5 LIMITATIONS

Our experiments are conducted on three popular datasets such as FMNIST, CIFAR-10, and ImageNette. As model complexity and dataset size grow, especially with higher class counts, adversarial training becomes increasingly difficult (Zhang et al., 2019). Maintaining both high accuracy and robustness in such settings presents a significant challenge. Additionally, while we explored several local and non-local smoothing filters, the choice of the optimal filter remains largely empirical and task-dependent.

## 6 CONCLUSION

In this paper, we explore the connection between model training strategies and quality of explanations, and propose a simple modification to adversarial training to improve the comprehensibility of saliency maps. Through a comprehensive study, we established that the quality of saliency maps is tied to the sensitivity of a model, with adversarially trained models producing sparser but unstable explanations. Incorporating local feature-map smoothing during adversarial training enhances stability and faithfulness without sacrificing sparsity. Our work underscores that meaningful and faithful explanations are tied to the model training strategy. By shedding light on the trade-offs between robustness of a model and saliency map quality, we advocate for the designing models that strike a balance between performance and saliency map comprehensibility.

---

[1]An Institutional Review Board (IRB) approval was granted by our institution prior to interviewing human subjects for our qualitative study.

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

## A  DATASET AND TRAINING

**FMNIST (Xiao et al., 2017):** The Fashion MNIST dataset consists of 28x28 pixel grayscale images of various clothing items and accessories. It contains a total of 70,000 images, divided into a training set of 60,000 examples and a test set of 10,000 examples. Similar to (Chalasani et al., 2020), we train a neural network consisting of two convolutional layers with 32 and 64 filters, respectively, each followed by 2x2 max-pooling and a fully connected layer of 1024. We use the Adam optimizer with a learning rate of 0.001, a batch size of 32 and 50 training epochs.

**CIFAR-10 (Krizhevsky et al., 2009):** CIFAR-10 consists of 60,000 32x32 pixel color images, with each image belonging to one of ten different classes. These classes include common objects and animals such as airplanes, automobiles, birds, cats, deer, dogs, frogs, horses, ships, and trucks. Similar to (Chalasani et al., 2020), we use a wide Residual Network (Zagoruyko & Komodakis, 2016) for training CIFAR-10 with the following hyperparameter settings: batch size=128, momentum optimizer with momentum = 0.9, and weight decay = 5e-4, training steps = 70000. We use an adaptive learning rate where the learning rate is set to 0.1 for the first 40000 steps, 0.01 for 40000-50000 steps, and 0.001 for the remaining steps. The wide residual network is trained with 28 layers and widen factor of 10.

**ImageNette (Howard, 2020):** ImageNette is a 10-class subset of ImageNet (Deng et al., 2009) with 9469 training images and 3925 test images. We use the 320-pixel resolution images (for the shortest side) and randomly resize and crop them to 224x224 pixels during training. We use the standard ResNet-18 model architecture for training on the dataset. We use Ranger optimizer (Wright, 2019) with an initial learning rate of 8e-03 and epsilon 1e-6. We train the models from scratch for 200 epochs and employ the early stopping criterion to select the best-performing model for evaluation.

### A.1  ADVERSARIAL TRAINING

Adversarial training (Goodfellow et al., 2015) is a machine learning technique that involves training a model in the presence of adversarial examples. Adversarial examples are inputs specifically designed to mislead or deceive the model, causing the model to make incorrect predictions. The goal of adversarial training is to improve the robustness and generalization of a model against such perturbed examples. To perform adversarial training, we generate adversarial examples that are produced from natural samples $\mathbf{x} \in R^d$ by adding a perturbation vector $\delta \in R^d$. The perturbation vector differs based on the type of attack employed. We use the PGD (Madry et al., 2018) attack to obtain adversarial perturbations. PGD is an iterative attack where the perturbation is computed multiple times with small steps. The hyper-parameters of PGD attack in our adversarial training: for FMNIST and CIFAR-10, $\epsilon \in \{0.01, 0.03, 0.06, 0.1\}$, attack step size = $\epsilon/10$, and number of iterations = 40; for ImageNette $\epsilon = 1/255$, step size = 0.00784 and number of iterations = 20. Other training hyperparameters are kept as explained in Appendix A.

## B  SMOOTHING FILTERS

A generic convolutional neural network with a feature map smoothing block is presented in Figure 14. The smoothing block consists of local or non-local filtering operations. All feature-map smoothed models are trained with the same hyper-parameter settings as explained in Appendix A. We use with the following filters in the paper:

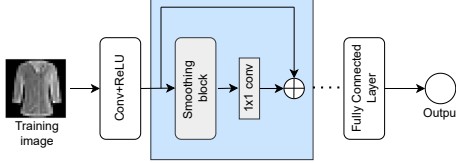

Figure 14: A generic convolutional neural network with a feature-map smoothing block.

## B.1 LOCAL SMOOTHING:

Local smoothing applies filtering operations to a neighborhood of a feature map. We use the following local smoothing filters in our approach:

- **Mean filter:** A mean filter, equivalent to an average pooling with a stride of 1, replaces each feature with the average of nearby features within a defined kernel. This smoothing effect reduces noise and enhances robustness to spatial variations. For an input feature map ($I$) of size $H$x$W$ and a $K$-sized kernel, the output feature map $O(u, v)$ is calculated using Eqn. 8:

$$O(u, v) = \frac{1}{K^2} \sum_{i=0}^{K-1} \sum_{j=0}^{K-1} I(u + i, v + j) \qquad (8)$$

  Here, $u$ and $v$ represent spatial coordinates in the output feature map, ranging from 0 to $H - K$ and 0 to $W - K$ respectively. $I(u + i, v + j)$ denotes the feature value at spatial location $(u + i, v + j)$ in the input feature map. This operation is applied independently to each channel of the input feature map.

- **Median filter:** A median filter, unlike a mean filter, computes the median value within a small sliding window over the feature map, given by Eqn. 9. This method also removes noise, making representations more robust. It also preserves edges and fine details as it selects the median value. Given an input feature map $I$ and a median filter window size $K$, the output feature map $O(u, v)$ is computed using Eqn. 9:

$$O(u, v) = median(I(u - \frac{K}{2} : u + \frac{K}{2}, v - \frac{K}{2} : v + \frac{K}{2}) \qquad (9)$$

  Here, $I(u - \frac{K}{2} : u + \frac{K}{2}, v - \frac{K}{2} : v + \frac{K}{2})$ represents the subset of the input feature around $(u, v)$ with a size of $K$x$K$. This operation is applied independently to each channel of the input feature map. Since median filters are non-linear and non-differentiable operations, this can pose challenges when training a neural network end-to-end. We utilize the approximation of the median filter available in Kornia eri (2020), which is differentiable.

- **Gaussian filter:** A Gaussian filter applies a smoothing effect to feature maps by convolving them with a Gaussian kernel, effectively reducing Gaussian noise. This process improves the signal-to-noise ratio and preserves edges better than mean filtering due to the Gaussian kernel giving more weight to nearby features while still considering distant feature contributions. The degree of smoothing can be adjusted by modifying the standard deviation ($\sigma$) of the Gaussian kernel. Given an input feature map $I$ and a Gaussian filter kernel $K$, the output feature map $O(u, v)$ is calculated with Eqn. 10:

$$O(u, v) = (I * K)(u, v) \qquad (10)$$

  Here, $*$ denotes 2D convolution. The Gaussian kernel $K$ is generated using a Gaussian function with a specific standard deviation $\sigma$, defined in Eqn. 11:

$$K(u, v) = \frac{1}{2\pi\sigma^2} e^{(-\frac{u^2 + v^2}{2\sigma^2})} \qquad (11)$$

  This operation is independently applied to each channel of the input feature map.

**Implementation:** We utilize the differentiable filters available in Kornia eri (2020). We use a 3x3 Kernel for mean, median, and Gaussian filtering. The standard deviation of the kernel for Gaussian filtering was computed as (0.3 * ((x.shape[3] - 1) * 0.5 - 1) + 0.8, 0.3 * ((x.shape[2] - 1) * 0.5 - 1) + 0.8) where x is the input image.

## B.2 NON-LOCAL SMOOTHING:

The non-local approach Buades et al. (2005) derives a smooth feature map $m$ from an input feature map $x$ by calculating a weighted average of features across all spatial positions within the set $\mathcal{L}$. Eqn. 12 shows the formulation where $f(x_i, x_j)$ is feature dependent weighting function and $\mathcal{C}(x)$ is a normalization function.

Table 2: Faithfulness evaluation of Vanilla Gradient (VG), Integrated Gradient (IG) & SmoothGrad (SG)

| | | FMNIST | | | | | | | CIFAR-10 | | | | | | | ImageNette | | | | | |
|---|---|---|---|---|---|---|---|---|---|---|---|---|---|---|---|---|---|---|---|---|---|
| | | N | A | M1 | M2 | G | E | NG | N | A | M1 | M2 | G | E | NG | N | A | M1 | M2 | G | E |
| VG | SIC | 0.67 | 0.68 | 0.70 | 0.67 | 0.67 | 0.68 | 0.67 | 0.26 | 0.69 | 0.65 | 0.67 | 0.68 | 0.64 | 0.67 | 0.62 | 0.66 | 0.68 | 0.70 | 0.70 | 0.74 | 0.73 |
| | AIC | 0.72 | 0.72 | 0.74 | 0.74 | 0.72 | 0.74 | 0.73 | 0.30 | 0.67 | 0.63 | 0.68 | 0.67 | 0.65 | 0.59 | 0.60 | 0.62 | 0.76 | 0.73 | 0.69 | 0.71 | 0.67 |
| | Faithfulness estimate | 0.45 | 0.82 | 0.86 | 0.73 | 0.86 | 0.72 | 0.81 | 0.07 | 0.16 | 0.13 | 0.19 | 0.14 | 0.17 | 0.16 | 0.07 | 0.32 | 0.38 | 0.40 | 0.34 | 0.36 | 0.42 |
| IG | SIC | 0.23 | 0.24 | 0.26 | 0.24 | 0.27 | 0.23 | 0.23 | 0.29 | 0.66 | 0.65 | 0.68 | 0.70 | 0.60 | 0.68 | 0.59 | 0.71 | 0.73 | 0.69 | 0.63 | 0.77 | 0.73 |
| | AIC | 0.28 | 0.33 | 0.31 | 0.33 | 0.35 | 0.28 | 0.35 | 0.31 | 0.66 | 0.68 | 0.64 | 0.66 | 0.68 | 0.58 | 0.65 | 0.75 | 0.78 | 0.74 | 0.68 | 0.76 | 0.75 |
| | Faithfulness estimate | 0.90 | 0.94 | 0.93 | 0.96 | 0.96 | 0.94 | 0.93 | 0.19 | 0.25 | 0.27 | 0.28 | 0.24 | 0.26 | 0.27 | 0.24 | 0.35 | 0.33 | 0.36 | 0.33 | 0.36 | 0.37 |
| SG | SIC | 0.41 | 0.53 | 0.54 | 0.52 | 0.52 | 0.43 | 0.52 | 0.26 | 0.55 | 0.52 | 0.53 | 0.62 | 0.57 | 0.62 | 0.59 | 0.77 | 0.72 | 0.79 | 0.67 | 0.81 | 0.78 |
| | AIC | 0.49 | 0.64 | 0.66 | 0.64 | 0.64 | 0.52 | 0.64 | 0.29 | 0.54 | 0.65 | 0.52 | 0.56 | 0.58 | 0.43 | 0.65 | 0.84 | 0.81 | 0.86 | 0.78 | 0.79 | 0.81 |
| | Faithfulness estimate | 0.86 | 0.90 | 0.90 | 0.92 | 0.90 | 0.82 | 0.90 | 0.33 | 0.56 | 0.55 | 0.59 | 0.54 | 0.56 | 0.56 | 0.72 | 0.77 | 0.75 | 0.76 | 0.73 | 0.66 | 0.71 |

$$m_i = \frac{1}{\mathcal{C}(x)} \sum_{\forall j \in \mathcal{L}} f(x_i, x_j).x_j \tag{12}$$

We consider the following forms of weighting function $f(.)$:

- **Non-local Gaussian Wang et al. (2018):** Eqn. 13 formulates the non-local gaussian function where $x_i^T x_j$ is the dot product similarity between the feature maps. The normalization function is set as $\mathcal{C}(x) = \sum_{\forall x} f(x_i, x_j)$.

$$f(x_i, x_j) = e^{(x_i^T x_j)} \tag{13}$$

- **Embedded Gaussian Wang et al. (2018):** This non-local mean computes similarity in embedding space by computing embedded versions of the feature map $x$. As shown in Eqn. 14, $\theta(x_i) = W_\theta x_i$ and $\eta(x_j) = W_\phi x_j$ are the two embeddings of feature map $x$, obtained after 1×1 convolution. The normalization function is set as $\mathcal{C}(x) = \sum_{\forall x} f(x_i, x_j)$.

$$f(x_i, x_j) = e^{(\theta(x_i)^T \eta(x_j))} \tag{14}$$

We use the open-source implementation of non-local means available in Github git (2018).

## C   FAITHFULNESS EVALUATION

In addition to the faithfulness evaluation using ROAD Rong et al. (2022), we evaluate faithfulness of explanations using faithfulness estimate (Alvarez Melis & Jaakkola, 2018), Softmax Information Curve (SIC) (Kapishnikov et al., 2019), and Accuracy Information Curve (AIC) (Kapishnikov et al., 2019). As presented in Table 2 shows that all robust models exhibit significantly higher faithfulness than their naturally trained counterparts, particularly on datasets like CIFAR-10 and ImageNette. This aligns with the findings of Shah et al. (2021), which showed that naturally trained models fail to capture the most discriminative features, often due to feature leakage.

However, while adversarial training appears to mitigate the feature leakage issue, and improves the faithfulness of explanations, the underlying mechanisms are still not fully understood. One hypothesis is that adversarial training encourages models to rely on more robust, generalizable features, which better reflect the decision-making process across adversarial and clean inputs. However, further research is needed to explore how adversarial training systematically reduces feature leakage and whether it can enhance the interpretability of other types of explanations, such as counterfactual explanations.

## D   EFFECT OF SMOOTHING FILTER

In Table 3, we present the results of various models on FMNIST, CIFAR-10 and ImageNette, with both natural (benign) and adversarial (robust) accuracy. Benign accuracy measures the model performance on benign (clean) test set, whereas robust accuracy evaluates how well the models detect

Table 3: Natural and Robust Accuracy of Various FMNIST, CIFAR-10, and ImageNette Models

| Dataset | Models/Accuracy | N | A | M1 | M1+A | M2 | M2+A | G | G+A | E | E+A | NG | NG+A |
|---------|-----------------|-----|------|------|------|------|------|------|------|------|------|-------|------|
| FMNIST | **Benign Accuracy** | 89.9 | 79.9 | 88.4 | 80.0 | 88.8 | 80.5 | 89.1 | 80.3 | 89.4 | 81.1 | 89.23 | 81.3 |
| | **Robust Accuracy** | 9.5 | 67.7 | 8.5 | 67.1 | 8.2 | 68.6 | 6.9 | 66.8 | 7.31 | 64.7 | 7.23 | 69.5 |
| CIFAR-10 | **Benign Accuracy** | 90.9 | 80.5 | 89.7 | 79.6 | 88.6 | 80.1 | 90.2 | 80.8 | 90.6 | 79.6 | 89.9 | 81.9 |
| | **Robust Accuracy** | 4.8 | 54.3 | 4.5 | 51.2 | 4.7 | 56.3 | 6.8 | 53.9 | 5.1 | 55.5 | 7.1 | 55.8 |
| ImageNette | **Benign Accuracy** | 96.3 | 70.8 | 93.3 | 58.8 | 90.9 | 55.3 | 95.5 | 51.6 | 88.4 | 60.8 | 86.3 | 58.4 |
| | **Robust Accuracy** | 1.6 | 12.2 | 1.2 | 6.5 | 2.3 | 14.3 | 3.7 | 13.5 | 3.1 | 13.9 | 2.5 | 18.9 |

Table 4: Result for adding smoothing block after second residual block

| | CIFAR-10 | | | | |
|---|---|---|---|---|---|
| | **M1** | **M2** | **G** | **E** | **NG** |
| **dG (higher the better)** | 0.178 | 0.185 | 0.176 | 0.190 | 0.191 |
| **dRIS (lower the better)** | -0.605 | -0.663 | -0.477 | -0.528 | -0.621 |
| **dROS (lower the better)** | 0.268 | 0.225 | 0.239 | 0.273 | 0.269 |
| **dRRS (lower the better)** | 0.464 | 0.445 | 0.462 | 0.453 | 0.475 |

adversarially perturbed samples. The robust models under evaluation are trained at $\epsilon = 0.1$ for FMNIST and CIFAR-10 and $\epsilon = 1/255$ for ImageNette. Evaluation is performed on a test-set consisting of adversarial samples created using PGD attack Madry et al. (2018) at $\epsilon = 0.1$ $l_\infty$ perturbation bound.

Across all datasets, applying smoothing filters alone did not result in significant changes in natural or robust accuracy ($\approx \pm 3\%$). The smoothing filters, when used without adversarial training, did not drastically improve robustness or reduce natural accuracy, indicating that their primary role may be in stabilizing feature maps without dramatically altering decision boundaries.

However, when smoothing filters were combined with adversarial training, robust accuracy improved for some filters, particularly in FMNIST and CIFAR-10, where models trained with adversarial samples and smoothing exhibited stronger defense against adversarial attacks. On the ImageNette dataset, we observed a notable drop in benign accuracy when smoothing filters were applied during adversarial training.

# E ABLATION STUDY: POSITION OF SMOOTHING FILTERS

In this section, we investigate how the placement of smoothing filters within the network affects the stability and sparsity of saliency maps. Specifically, we consider different positions for inserting the smoothing filters in a CIFAR-10 network and report the results in Tables 4and 5 for Vanilla Gradient. This CIFAR-10 Residual Network consists of three residual blocks. We add smoothing filters after second residual block in Table 4 and after third residual block in Table 5. In Table 1, smoothing filters are added after first residual block.

Across all residual blocks, the sparsity gain remains consistent between 0.176 to 0.192; however, when smoothing filter is added after third residual block, there is a slight improvement in the sparsity. Smoothing after the first block consistently yields better results in stability. Hence, to strike a balance between stability and sparsity, we place the smoothing block after the first residual block.

Table 5: Result for adding smoothing block after third residual block

| | CIFAR-10 | | | | |
|---|---|---|---|---|---|
| | **M1** | **M2** | **G** | **E** | **NG** |
| **dG (higher the better)** | 0.185 | 0.180 | 0.187 | 0.191 | 0.192 |
| **dRIS (lower the better)** | -0.599 | -0.670 | -0.470 | -0.517 | -0.612 |
| **dROS (lower the better)** | 0.271 | 0.221 | 0.235 | 0.276 | 0.261 |
| **dRRS (lower the better)** | 0.470 | 0.429 | 0.468 | 0.446 | 0.473 |

Table 6: Sparsity and Stability Evaluations for VG, IG, and SG. Here, ↑ and ↓ indicate higher and lower values are better.

| | Vanilla Gradient (VG) | | | | | | Integrated Gradient (IG) | | | | | | SmoothGrad (SG) | | | | | |
|---|---|---|---|---|---|---|---|---|---|---|---|---|---|---|---|---|---|---|
| | A | M1 | M2 | G | E | NG | A | M1 | M2 | G | E | NG | A | M1 | M2 | G | E | NG |
| **dG** ↑ | 0.10 | 0.10 | 0.10 | 0.10 | 0.11 | 0.09 | 0.02 | 0.03 | 0.02 | 0.01 | 0.01 | 0.02 | 0.08 | 0.08 | 0.08 | 0.08 | 0.08 | 0.08 |
| **dRIS** ↓ | -0.30 | -0.40 | -0.35 | -0.39 | -0.39 | -0.42 | -0.29 | -0.62 | -0.74 | -0.60 | -0.84 | -0.81 | -0.33 | -0.36 | -0.46 | -0.10 | -0.49 | -0.52 |
| **dROS** ↓ | -0.24 | -0.31 | -0.26 | -0.30 | -0.30 | -0.32 | -0.13 | -0.22 | -0.52 | -0.24 | -0.52 | -0.56 | -0.42 | -0.50 | -0.49 | -0.40 | -0.47 | -0.53 |
| **dRRS** ↓ | 0.28 | 0.21 | 0.25 | 0.19 | 0.19 | 0.18 | 0.24 | 0.17 | -0.25 | 0.04 | -0.35 | -0.24 | 0.06 | 0.03 | 0.02 | 0.05 | 0.01 | -0.09 |

# F    ADDITIONAL EXPERIMENTS

In this section, we demonstrate the effects of robust training strategy on saliency map quality for a different network, VGG16 Simonyan & Zisserman (2015) on CIFAR-10. We train a VGG-16 convolutional neural network for 120 epochs using stochastic gradient descent (SGD) with momentum, a learning rate of 0.1, and weight decay of 5e-4. The model consists of five convolutional blocks with batch normalization, ReLU activations, max-pooling layers, and a fully connected classifier. The training utilizes a learning rate scheduler, which reduces the learning rate by a factor of 0.1 every 30 epochs. For adversarial training, we use the same hyperparameter (PGD attack at $\epsilon = 0.1$). The hyperparameters for smoothing blocks are also kept as discussed before. Similar to previous sections, we train following models for VGG network: naturally-trained (N), adversarially-trained (A), adversarial training with mean-filter smoothing (M1), adversarial training with median-filter smoothing (M2), adversarial training with Gaussian-filter smoothing (G), adversarial training with embedded filter smoothing (E), and adversarial training with non-local gaussian smoothing (NG). .

Next to evaluate sparsity, and stability, for each model, we compute explanations using Vanilla Gradient (VG), Integrated Gradient (IG), and SmoothGrad (SG), and then compute its sparseness using Gini index (G) (Chalasani et al., 2020), and its stability using relative input stability (RIS), relative output stability (ROS) and relative representation stability (RRS) (Agarwal et al., 2022). Similar to Chalasini et al. (Chalasani et al., 2020), we compare the sparsity and stability improvement of saliency maps with respect to the naturally trained model (N). Specifically, for a given training method (M), we compute the following metrics that quantify the improvement in sparseness (dG), relative input stability (dRIS), relative output stability (dROS), and relative representation stability (dRRS) of the explanation method $\phi(.) \in \{VG, IG, SG\}$:

$$dG[\phi(\mathbf{x})] = G^M[\phi(\mathbf{x})] - G^N[\phi(\mathbf{x})] \tag{15}$$

$$dRIS[\phi(\mathbf{x})] = RIS^M[\phi(\mathbf{x})] - RIS^N[\phi(\mathbf{x})] \tag{16}$$

$$dROS[\phi(\mathbf{x})] = ROS^M[\phi(\mathbf{x})] - ROS^N[\phi(\mathbf{x})] \tag{17}$$

$$dRRS[\phi(\mathbf{x})] = RRS^M[\phi(\mathbf{x})] - RRS^N[\phi(\mathbf{x})] \tag{18}$$

Table 6 shows the results of sparsity and stability evaluation of saliency maps generated by Vanilla Gradient (VG), Integrated Gradient (IG), and SmoothGrad (SG) across a variety of models in VGG network. We can observe that all explanation methods show positive dG values across all models, indicating that the saliency maps become sparser when used with robust, adversarially trained VGG models. The sparsity gain, however, remains relatively stable across models, with only slight variations. This suggests that while robust training introduces sparsity, the choice of smoothing filter does not significantly impact the sparsity of explanations.

In terms of input and output stability ($dRIS$ and $dROS$), we observe that models enhanced with smoothing filters (M1, M2, G, E, NG) consistently exhibit better stability compared to the adversarially trained baseline (A). This is particularly pronounced in the IG and SG methods, where stability improvements are more significant. The introduction of smoothing filters, such as median and Gaussian, mitigates the instability of explanations seen in the baseline model, resulting in more reliable and interpretable saliency maps.

## G  CONDITIONS AFFECTING THE TIGHTNESS OF STABILITY BOUNDS

The stability bounds presented in Section 3.1 serve as indicators of the relationship between model sensitivity and attribution stability. However, these bounds are inherently approximate and depend on several factors, including model architecture, the input data distribution, and the type of perturbation applied. Here, we discuss some conditions under which these bounds may become tighter or looser.

1. **Model Nonlinearity and Activation Function:** The nonlinearity of the model, particularly the choice of activation function $H$, influences the bounds' tightness. For activation functions with bounded gradients, such as sigmoid or tanh, the change in $H'(\langle \mathbf{w}, \mathbf{x} \rangle)$ is limited, leading to more consistent attributions across small perturbations and therefore tighter stability bounds. Specifically, for sigmoid, $H(z) = \frac{1}{1+e^{-z}}$ and $H'(z) = H(z)(1 - H(z))$, both of which remain bounded as $H(z)$ approaches 0 or 1. Conversely, for ReLU activation, $H(z) = \max(0, z)$ with $H'(z) = 1$ when $z > 0$ and 0 otherwise, the gradient can change abruptly across input perturbations. Thus, for perturbations where $\mathbf{x}$ is shifted across the activation boundary, $H'(\langle \mathbf{w}, \mathbf{x} \rangle)$ may vary significantly, producing looser bounds.

2. **Magnitude and Type of Input Perturbations:** The type and scale of input perturbations can also impact bound tightness. For small perturbations, such as Gaussian noise with $\mathbf{n} \sim \mathcal{N}(0, \sigma^2)$, the output change is typically small, and stability bounds remain tight. However, larger perturbations, such as stronger adversarial attack, often result in more significant output shifts $|F(\mathbf{x}') - F(\mathbf{x})|$, leading to looser bounds.

3. **Smoothness of Model Parameters:** Weight regularization techniques, such as weight decay, result in smoother gradients, reducing the sensitivity of $F(\mathbf{x})$ to input changes. For instance, regularized models with smaller gradient norms tend to have tighter stability bounds as $H'(\langle \mathbf{w}, \mathbf{x} \rangle) \cdot \mathbf{w}$ varies less across the input space. Consequently, the bounds for VG, SG, and IG become tighter, as regularization reduces model sensitivity.

4. **Dataset-Specific Characteristics:** Datasets with high intraclass variability introduce more variable responses to perturbations, increasing $|F(\mathbf{x}') - F(\mathbf{x})|$. As a result, stability bounds may become looser due to the variability in $F(\mathbf{x})$ across samples.

## H  STUDY ON RECEPTIVE FIELD EXPANSION

To measure the receptive field effect in the smoothing block, we conduct an additional experiment on CIFAR-10 where we modify the feature smoothing block so that it performs only a convolution (identify or randomly initialized). This modified setup ensures that there is only an expansion of the receptive field without filtering operations and it can provide a baseline study to analyze the effect of receptive field expansion on its own. Table 7 shows the results for Vanilla Gradient (VG) when compared with the best performing model.

Table 7: Sparsity and Stability evaluation for Vanilla Gradients. *Here, M2: adversarial training with median smoothing, Identity: adversarial training with feature smoothing block consisting of identify convolution but no smoothing filter and Random: adversarial training with feature smoothing block consisting of randomly initialized convolution but no smoothing filters*

| Models | M2 | Identity | Random |
|---|---|---|---|
| Sparsity (dG) (hgiher is better) | 0.18 | 0.16 | 0.15 |
| Relative input stability (dRIS) (lower is better) | -0.68 | -0.41 | -0.36 |
| Relative output stability (dROS) (lower is better) | 0.21 | 0.07 | 0.06 |
| Relative representation stability (dRRS) (lower is better) | 0.43 | 0.41 | 0.43 |

The results in the table show that:

1. The 'M2' model still achieves the best sparsity, indicating that the smoothing operation in addition to the convolutional operation helps the model to learn a smaller number of discriminative features.

2. The 'M2' model performs significantly better in input stability. This indicates that smoothing filters provide stability in saliency maps with respect to input.

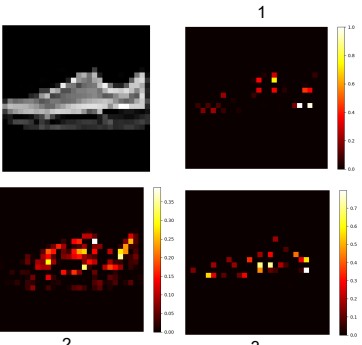

Figure 15: A test image and corresponding saliency maps for models used in the survey.

Table 8: Wilcoxon and ANOVA test results on the survey

|  | Wilcoxon (p-value) | | | one-way ANOVA | |
|---|---|---|---|---|---|
|  | 1 vs 2 | 2 vs 3 | 1 vs 3 | F-stat | p value |
| **Sufficiency** | 9.79E-41 | 4.26E-14 | 3.71E-27 | 200.38 | 7.82E-72 |
| **Trust** | 5.56E-39 | 3.24E-11 | 3.89E-24 | 193.86 | 6.58E-70 |

3. Interestingly, the 'M2' model does not achieve the best score in output stability. This suggests that while smoothing helps in stabilizing attributions with respect to inputs and internal representations, it might not directly translate to stability at the model's output layer. The expanded receptive field introduced by the identity or random convolutions likely contributes to this improvement.

4. The 'Identity' model achieves the best representation stability but only marginally outperforming 'M2'.

Overall, the inclusion of smoothing operations still provides a competitive advantage in improving the quality of saliency maps with respect to sparsity, input stability and representation stability.

## I  ON QUALITATIVE STUDY

Our quantitative studies show that robust models produce sparse explanations at the expense of stability. The inclusion of local feature-map smoothing enhances the stability without a significant reduction in sparsity, striking a balance between sparsity and stability in the resulting saliency maps. In this section, we present our analysis of how effectively end-users understand the saliency maps of different model training strategies based on the level of sparsity. We conducted an experiment involving human subjects where participants were asked to interpret the saliency maps from two image datasets: FMNIST and CIFAR-10. Their responses were recorded and assessed using the Hoffman et al. satisfaction scale Hoffman et al. (2018)[2].

**Survey Methodology:** Comprehension of explanations and their impact is known to be significantly influenced by the expertise of its end-users Wang & Yin (2021). Hence, we interviewed 65 graduate students (Ph.D./Masters) with a minimum of one year of experience in computer vision. The main goal was to determine if the information conveyed by saliency maps was sufficient to understand and trust the underlying model behavior.

We initiated our study by explaining how to read the saliency maps in the context of image classification tasks, emphasizing the meaning behind different pixel colors. Once participants understood the concept of the saliency maps, we showed them a set of ten images, each accompanied by its respective saliency maps generated from three distinct models: a naturally trained model, an adversarially trained model, and an adversarially trained model with feature map smoothing (median filter), resulting in a total of 60 image-saliency map pairs. To avoid bias, the order of the saliency

---

[2]An Institutional Review Board (IRB) approval was granted by our institution prior to interviewing human subjects for our qualitative study.

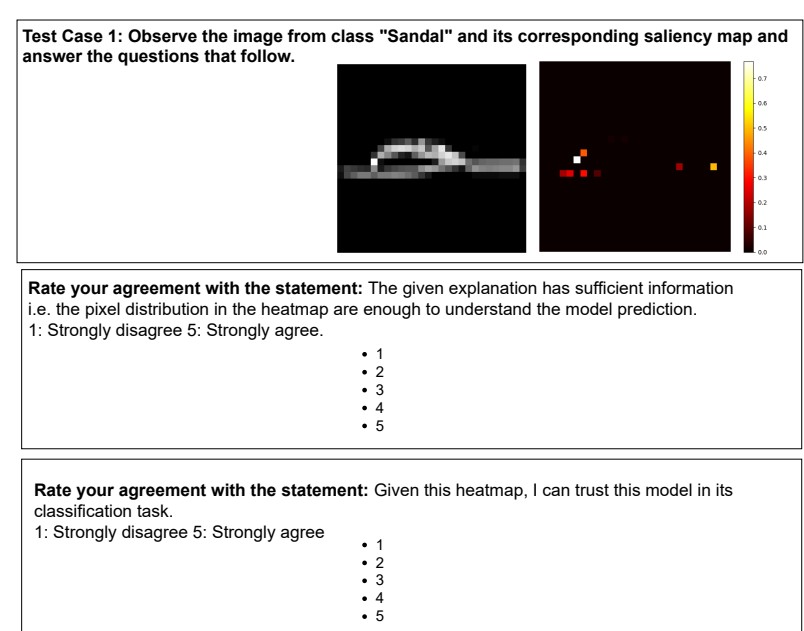

Figure 16: A sample of question from the survey.

maps presentation was randomized, and the participants did not know the model that produced each saliency map. We then asked them to complete a survey based on the Hoffman et al. satisfaction scale Hoffman et al. (2023). This survey included two questions for each image-saliency map pair: 1) "Does the given explanation have sufficient information?" and 2) "Given this heatmap, do you trust the model's classification?". Participants rated their agreement on a scale from 1 (strongly disagree) to 5 (strongly agree) (See Figure 16 for a sample). After reviewing the saliency maps from each model individually, we showed the participants saliency maps from all three models side-by-side, and asked them to select the most comprehensible explanation (as shown in Figure 15). We also collected free-text answers by asking them to explain the reason behind their selection.

In Table 8, we present the results of two statistical tests, the Wilcoxon signed-rank test Woolson (2007) and one-way ANOVA Cuevas et al. (2004), for inspecting if there is a significant difference between the trust and sufficiency metrics of the three different models, (1) a naturally trained model, (2) an adversarially-trained model, and (3) an adversarial-trained feature-map smoothed model. The extremely small p-values (less than 0.001) suggest that there are significant differences between both "sufficiency" and "trust" metrics of saliency maps across the three models.

## J    RELATIONSHIP BETWEEN ATTRIBUTION STABILITY AND MODEL SENSITIVITY

Consider a single-layer DNN with the form $F(\mathbf{x}) = H(\langle \mathbf{w}, \mathbf{x} \rangle)$, where $H$ is a differentiable scalar-valued activation function (e.g., sigmoid), $\langle \mathbf{w}, \mathbf{x} \rangle$ is the dot product between the weight vector $\mathbf{w}$ and input $\mathbf{x} \in \mathbb{R}^d$.

### J.1    RELATIONSHIP FOR VANILLA GRADIENT (VG)SIMONYAN ET AL. (2014)

Let $\mathbf{x} \in R^d$ denote an input image. The Vanilla Gradient (VG) explanation for a model $F$ is computed as,

$$VG(\mathbf{x}) = \frac{\partial F_c(\mathbf{x})}{\partial \mathbf{x}} \tag{19}$$

For a single-layer DNN with the form $F(\mathbf{x}) = H(\langle \mathbf{w}, \mathbf{x} \rangle)$, where $H$ is a differentiable scalar-valued activation function, $\langle \mathbf{w}, \mathbf{x} \rangle$ is the dot product between the weight vector $\mathbf{w}$ and input $\mathbf{x} \in \mathbb{R}^d$, the VG can be computed by applying the chain rule as follows:

$$VG(\mathbf{x}) = \frac{\partial H(\langle \mathbf{w}, \mathbf{x} \rangle)}{\partial \langle \mathbf{w}, \mathbf{x} \rangle} \cdot \frac{\partial \langle \mathbf{w}, \mathbf{x} \rangle}{\partial \mathbf{x}} = H'(\langle \mathbf{w}, \mathbf{x} \rangle).\mathbf{w} \tag{20}$$

Here, $H'(\langle \mathbf{w}, \mathbf{x} \rangle)$ is the gradient of activation function $H$ with respect to the $\langle \mathbf{w}, \mathbf{x} \rangle$. Let $z = \langle \mathbf{w}, \mathbf{x} \rangle$ and $H(z) = \frac{1}{1+exp(-z)}$ be a sigmoid activation function then,

$$H'(z) = \frac{exp(-z)}{(1 + exp(-z))^2} \tag{21}$$
$$= \frac{1}{1 + exp(-z)}(1 - \frac{1}{1 + exp(-z)})$$
$$= H(z)(1 - H(z))$$

Then, the VG attribution for an input $\mathbf{x}$ is given by

$$VG^F(\mathbf{x}) = H(\langle \mathbf{w}, \mathbf{x} \rangle)(1 - H(\langle \mathbf{w}, \mathbf{x} \rangle)).\mathbf{w} \tag{22}$$

Now consider $\mathbf{x}' \in \mathcal{N}_\mathbf{x}$ is a noisy version of input image $\mathbf{x}$ where $\mathcal{N}_\mathbf{x}$ indicates a neighborhood of inputs $\mathbf{x}$ where the model prediction is locally consistent. Then, the VG attribution for an input $\mathbf{x}'$ is given by

$$VG^F(\mathbf{x}') = H(\langle \mathbf{w}, \mathbf{x}' \rangle)(1 - H(\langle \mathbf{w}, \mathbf{x}' \rangle)).\mathbf{w} \tag{23}$$

The stability of the VG attribution is computed as the norm of the difference between the attribution of the original image and its noisy counterpart and can be expressed as

$$\Delta = ||VG^F(\mathbf{x}') - VG^F(\mathbf{x})||_1 \tag{24}$$

Substituting the expressions for $VG^F(\mathbf{x})$ and $VG^F(\mathbf{x}')$, and simplifying, we obtain

$$\Delta = ||VG^F(\mathbf{x}') - VG^F(\mathbf{x})||_1$$
$$= ||H(\langle \mathbf{w}, \mathbf{x}' \rangle)(1 - H(\langle \mathbf{w}, \mathbf{x}' \rangle))\mathbf{w} - H(<\mathbf{w}, \mathbf{x}>)(1 - H(\langle \mathbf{w}, \mathbf{x} \rangle)).\mathbf{w}||_1$$
$$= ||\Big(H(\langle \mathbf{w}, \mathbf{x}' \rangle)(1 - H(\langle \mathbf{w}, \mathbf{x}' \rangle)) - H(\langle \mathbf{w}, \mathbf{x} \rangle)(1 - H(\langle \mathbf{w}, \mathbf{x} \rangle))\Big)\mathbf{w}||_1$$
$$= ||\Big(F(\mathbf{x}')(1 - F(\mathbf{x}')) - F(\mathbf{x})(1 - F(\mathbf{x}))\Big)\mathbf{w}||_1$$
$$= ||\Big((F(\mathbf{x}') - F(\mathbf{x}))(1 - F(\mathbf{x}') - F(\mathbf{x}))\Big)\mathbf{w}||_1$$

$$\tag{25}$$

Bounding this by the magnitude of the change in model prediction,

$$\Delta \le ||\Big(F(\mathbf{x}') - F(\mathbf{x})\Big)\mathbf{w}||_1$$
$$\Delta \le ||F(\mathbf{x}') - F(\mathbf{x})||_1.||\mathbf{w}||_1 \tag{26}$$

Assuming $\mathbf{w}$ to be constant for a given model, the stability of the VG attribution is a direct result of the sensitivity of the model $||F(\mathbf{x}') - F(\mathbf{x})||$.

## J.2 RELATIONSHIP FOR INTEGRATED GRADIENT (IG) SUNDARARAJAN ET AL. (2017)

The feature attribution score computed by Integrated Gradient (IG) for feature $i$ of input image $\mathbf{x} \in R^d$ with baseline $\mathbf{u}$, model $F$ is given by:

$$IG_i^F(\mathbf{x}, \mathbf{u}) = (x_i - u_i). \int_{\alpha=0}^{1} \partial_i F(\mathbf{u} + \alpha(\mathbf{x} - \mathbf{u})) \partial \alpha \tag{27}$$

For an input image $\mathbf{x}$, IG returns a vector $IG^F(\mathbf{x}, \mathbf{u}) \in R^d$ with scores that quantify the contribution of $x_i$ to the model prediction $F(\mathbf{x})$. For a single layer network $F(\mathbf{x}) = H(\langle \mathbf{w}, \mathbf{x} \rangle)$ where $H$ is a differentiable scalar-valued function and $\langle \mathbf{w}, \mathbf{x} \rangle$ is the dot product between the weight vector $\mathbf{w}$ and input $\mathbf{x} \in R^d$, IG attribution has a closed form expression Chalasani et al. (2020).

For given $\mathbf{x}$, $\mathbf{u}$ and $\alpha$, let us consider $\mathbf{v} = \mathbf{u} + \alpha(\mathbf{x} - \mathbf{u})$. If the single-layer network is represented as $F(\mathbf{x}) = H(\langle \mathbf{w}, \mathbf{x} \rangle)$ where $H$ is a differentiable scalar-valued function, $\partial_i F(\mathbf{v})$ can be computed as:

$$\begin{aligned}
\partial_i F(\mathbf{v}) &= \frac{\partial F(\mathbf{v})}{v_i} \\
&= \frac{\partial H(\langle \mathbf{w}, \mathbf{v} \rangle)}{\partial v_i} \\
&= H'(z) \frac{\partial \langle \mathbf{w}, \mathbf{v} \rangle}{\partial v_i} \\
&= w_i H'(z)
\end{aligned} \tag{28}$$

Here, $H'(z)$ is the gradient of the activation $H(z)$ where $z = \langle \mathbf{w}, \mathbf{v} \rangle$. To compute $\frac{\partial F(\mathbf{v})}{\partial \alpha}$:

$$\frac{\partial F(\mathbf{v})}{\partial \alpha} = \sum_{i=1}^{d} \left( \frac{\partial F(\mathbf{v})}{\partial v_i} \frac{\partial v_i}{\partial \alpha} \right) \tag{29}$$

We can substitute value of $\frac{\partial v_i}{\partial \alpha} = (x_i - u_i)$ and $\partial_i F(\mathbf{v})$ from Eq. 28 to Eq. 29.

$$\begin{aligned}
\frac{\partial F(\mathbf{v})}{\partial \alpha} &= \sum_{i=1}^{d} [w_i H'(z)(x_i - u_i)] \\
&= \langle \mathbf{x} - \mathbf{u}, \mathbf{w} \rangle H'(z)
\end{aligned} \tag{30}$$

This gives:
$$dF(\mathbf{v}) = \langle \mathbf{x} - \mathbf{u}, \mathbf{w} \rangle H'(z) \partial \alpha \tag{31}$$

Since $\langle \mathbf{x} - \mathbf{u}, \mathbf{w} \rangle$ is scalar,

$$H'(z) \partial \alpha = \frac{dF(\mathbf{v})}{\langle \mathbf{x} - \mathbf{u}, \mathbf{w} \rangle} \tag{32}$$

Eq. 32 can be used to rewrite the integral in the definition of $IG_i^F(\mathbf{x})$ in Eq. 27,

$$\begin{aligned}
\int_{\alpha=0}^{1} \partial_i F(\mathbf{v}) \partial \alpha &= \int_{\alpha=0}^{1} w_i H'(z) \partial z \quad \text{[From Eqn. 28]} \\
&= \int_{\alpha=0}^{1} w_i \frac{dF(\mathbf{v})}{\langle \mathbf{x} - \mathbf{u}, \mathbf{w} \rangle} \\
&= \frac{w_i}{\langle \mathbf{x} - \mathbf{u}, \mathbf{w} \rangle} \int_{\alpha=0}^{1} dF(\mathbf{v}) \\
&= \frac{w_i}{\langle \mathbf{x} - \mathbf{u}, \mathbf{w} \rangle} [F(\mathbf{x}) - F(\mathbf{u})]
\end{aligned} \tag{33}$$

Hence, we obtain the closed form for Integrated Gradient from its definition in Eqn. 27 as

$$IG_i^F(\mathbf{x}, \mathbf{u}) = [F(\mathbf{x}) - F(\mathbf{u})]\frac{(x_i - u_i)w_i}{\langle \mathbf{x} - \mathbf{u}, \mathbf{w} \rangle}$$

$$IG^F(\mathbf{x}, \mathbf{u}) = [F(\mathbf{x}) - F(\mathbf{u})]\frac{(\mathbf{x} - \mathbf{u}) \odot \mathbf{w}}{\langle \mathbf{x} - \mathbf{u}, \mathbf{w} \rangle} \tag{34}$$

Here, $\odot$ is the entry-wise product of two vectors.

Now consider $\mathbf{x}' \in \mathcal{N}_{\mathbf{x}}$ is a noisy version of input image $\mathbf{x}$ where $\mathcal{N}_{\mathbf{x}}$ indicates a neighborhood of inputs $\mathbf{x}$ where the model prediction is locally consistent. The stability of the IG attribution can be computed using Eqn. 35.

$$\Delta = ||IG^F(\mathbf{x}', \mathbf{u}) - IG^F(\mathbf{x}, \mathbf{u})||_1 \tag{35}$$

This is equivalent to,

$$\begin{aligned}\Delta &\approx ||IG^F(\mathbf{x}', \mathbf{x})||_1 \\ &= \left\lVert [F(\mathbf{x}') - F(\mathbf{x})]\frac{(\mathbf{x}' - \mathbf{x}) \odot \mathbf{w}}{\langle \mathbf{x}' - \mathbf{x}, \mathbf{w} \rangle} \right\rVert_1 \\ &= \left\lVert [F(\mathbf{x}') - F(\mathbf{x})]\frac{\Delta_x \odot \mathbf{w}}{\langle \Delta_x, \mathbf{w} \rangle} \right\rVert_1 \end{aligned} \tag{36}$$

Assuming $\mathbf{w}$ to be constant for a given model, we can conclude from Eqn. 36 that the sensitivity of the IG attribution is a direct result of the sensitivity of the model $||F(\mathbf{x}') - F(\mathbf{x})||$.

### J.3 RELATIONSHIP FOR SMOOTHGRAD (SG) SMILKOV ET AL. (2017)

To compute SmoothGrad (SG) (Smilkov et al., 2017), we introduce Gaussian noise $\mathbf{n} \sim \mathcal{N}(0, \sigma^2)$ to the input $\mathbf{x}$ and compute the input-gradient for multiple noisy samples $\mathbf{x}_k = \mathbf{x} + \mathbf{n}_k$ for $k = 1, \ldots, N$, where $N$ is the number of noise samples.

$$SG(\mathbf{x}) = \frac{1}{N}\sum_{k=1}^{N}\frac{\partial F(\mathbf{x}_k)}{\partial \mathbf{x}_k} \tag{37}$$

SG explanation is then obtained by averaging the explanations. Since SG is a simple averaging of Vanilla Gradient, the relationship for SG follows from relationship of VG, as shown in Section J.1.

## K EVALUATION METRICS

Below, we discuss evaluation metrics used in our experiments.

### K.1 SPARSITY CHALASANI ET AL. (2020)

We measure the sparsity of the attribution vector $\phi(\mathbf{x})$ by computing its Gini index, available in Quantus Hedström et al. (2023). Given a vector of attribution $\phi(\mathbf{x}) \in R^d$, the absolute of the vector is first sorted in non-decreasing order, and the Gini index is computed using Eqn. 38.

$$G(\phi(\mathbf{x})) = 1 - 2\sum_{k=1}^{d}\frac{\phi(\mathbf{x})_{(k)}}{||\phi(\mathbf{x})||_1}\frac{d - k + 0.5}{d} \tag{38}$$

The formula calculates a weighted sum of fractions, where each fraction represents the contribution of the k-th largest element to the overall sparsity. The formula assigns greater weight to larger

elements and smaller weight to smaller elements. The Gini Index values lie in between $[0, 1]$; A value of 1 indicates perfect sparsity, where only one element in the vector $\phi_i(\mathbf{x}) > 0$. The sparsity is zero if all the vectors are equal to some positive value.

## K.2 STABILITY AGARWAL ET AL. (2022)

The stability metric measures how similar explanations are for similar inputs. Relative input stability (given by Eqn. 39) is measured as the difference between two attribution vectors $\phi(\mathbf{x})$ and $\phi(\mathbf{x}')$ with respect to the difference between the two inputs $\mathbf{x}$ and $\mathbf{x}'$. $\mathbf{x}'$ is computed by perturbing $\mathbf{x}$. A lower RIS value shows that explanations are similar for similar inputs.

$$RIS = max_{\mathbf{x}'} \frac{||\frac{\phi(\mathbf{x})-\phi(\mathbf{x}')}{\phi(\mathbf{x})}||}{max(||\frac{\mathbf{x}-\mathbf{x}'}{\mathbf{x}}||_p, \epsilon_{min})}$$
$$\forall \mathbf{x}' \ s.t. \ \mathbf{x}' \in \mathcal{N}_{\mathbf{x}}; \hat{y}_{\mathbf{x}} = \hat{y}_{\mathbf{x}'} \tag{39}$$

Relative input stability only measures the difference in input space and does not measure whether there was a change in the logic path of a network for a perturbed input. Relative representation stability (given by Eqn. 40) uses the internal representation of the model $(a(.))$ to compute the stability.

$$RRS = max_{\mathbf{x}'} \frac{||\frac{\phi(\mathbf{x})-\phi(\mathbf{x}')}{\phi(\mathbf{x})}||}{max(||a(\mathbf{x}) - a(\mathbf{x}')||_p, \epsilon_{min})}$$
$$\forall \mathbf{x}' \ s.t. \ \mathbf{x}' \in \mathcal{N}_{\mathbf{x}}; \hat{y}_{\mathbf{x}} = \hat{y}_{\mathbf{x}'} \tag{40}$$

Relative output stability (given by Eqn. 41) measures the difference between two attribution vectors $\phi(\mathbf{x})$ and $\phi(\mathbf{x}')$ with respect to the difference between the model logits for two inputs $z(\mathbf{x})$ and $z(\mathbf{x}')$ when $\mathbf{x}$ is perturbed to produce $\mathbf{x}'$. A lower ROS value shows that explanations are similar for similar inputs.

$$ROS = max_{\mathbf{x}'} \frac{||\frac{\phi(\mathbf{x})-\phi(\mathbf{x}')}{\phi(\mathbf{x})}||}{max(||z(\mathbf{x}) - z(\mathbf{x}')||_p, \epsilon_{min})}$$
$$\forall \mathbf{x}' \ s.t. \ \mathbf{x}' \in \mathcal{N}_{\mathbf{x}}; \hat{y}_{\mathbf{x}} = \hat{y}_{\mathbf{x}'} \tag{41}$$

$\mathcal{N}_{\mathbf{x}}$ in Eqn. 39, Eqn. 40 and Eqn. 41 indicates a neighborhood of inputs $\mathbf{x}'$ similar to $\mathbf{x}$. We use the implementation of the stability metrics available in Quantus Hedström et al. (2023).

## K.3 FAITHFULNESS

**Faithfulness estimate Alvarez Melis & Jaakkola (2018):** The faithfulness metric measures the influence of attributed features on model prediction. If the features attributed to an explanation method truly capture the model behavior, the influence should be high. Influence is measured with a correlation metric where a given image is iteratively modified to compute the correlation between the sum of attributions and the difference in model prediction. We use the implementation of the faithfulness estimate available in Quantus Hedström et al. (2023).

**Performance Information Curves (PIC) Kapishnikov et al. (2019):** Performance Information Curves (PIC) is analogous to the area under Receive Operating Characteristics (ROC) curves, proposed by Kapishnikov et al., to measure the quality of saliency maps. There are two variants of PIC: Area Under Softmax Information Curve (SIC) and Area Under Accuracy Information Curve (AIC). To measure PIC, we take a blurred version of a given image and then unblur the pixels by adding features that are deemed important by an attribution method. We measure the entropy of the unblurred image and model performance and then map the model performance result as a

function of the entropy. The two variants of the PIC, AIC, and SIC, differ in the model performance metric used to compute the area under the curve. AIC uses the accuracy of images and SIC uses the proportion of the softmax. We use the implementation shared by the authors of the original paper PAIR (2021).

### K.4 ROAD: REMOVE AND DEBIAS RONG ET AL. (2022)

ROAD measures the accuracy of a model on the provided test set at each step of an iterative process of removing $k$ most important pixels. Removal of pixels is done with a noisy linear imputation to avoid out-of-distribution samples.

We use the MoRF (Most Relevant First) removal strategy implementation of the ROAD evaluation available in Quantus Hedström et al. (2023). Given a network $F$ and an input sample, an attribution method assigns an importance value to each input feature for the sample. The features are then ordered in decreasing order of importance for MoRF. At each iteration, $k$ most important features are removed and the model accuracy is measured. We set $k = 5$ in our experiments. We prefer a sharper drop in accuracy for a better explanation.

We use ROAD instead of Insertion/Deletion Petsiuk et al. (2018) or ROAR Hooker et al. (2019) because Insertion/Deletion introduces artifacts and results in a distribution shift of perturbed inputs, and ROAR requires an expensive model retraining.

### K.5 SIMILARITY ADEBAYO ET AL. (2018)

Similarity measures the structural similarity between saliency maps of original and perturbed samples, given the same model prediction Adebayo et al. (2018). We measure the similarity of saliency maps using the structural similarity index (SSIM). For each image x, we add Gaussian noise ($\mathcal{N}(0, \sigma)$) and generate its noisy version x' such that the model prediction is consistent. We then compute the saliency map of x and x' and measure the structural similarity between the maps.

## L  ADDITIONAL VISUALIZATION

We provide additional visualizations on Vanilla Gradient (VG) in Figures 17, 18 and 19 for various models: naturally-trained (N), adversarially-trained (A), adversarial training with mean-filter smoothing (M1), adversarial training with median-filter smoothing (M2), adversarial training with Gaussian-filter smoothing (G), adversarial training with embedded filter smoothing (E), and adversarial training with non-local gaussian smoothing (NG). We can observe that saliency maps from the adversarial models (A) are sparser than the naturally trained model (N). Adversarially trained models with local feature map smoothed models (M1, M2, G) reduce the sparsity to improve stability. The use of non-local smoothing filters (E and NG) increases the sparsity further.

We plot the saliency maps using Integrated Gradient (IG) for various models in Figures 20, 21 and 22. As illustrated, IG produces more fine-grained saliency maps than Vanilla Gradient even with a naturally trained model. Robust models increase the sparsity of such saliency maps, compromising stability. Adding local filters like median during adversarial training reduces sparsity to enhance stability.

We provide illustrations for SmoothGrad (SG) in Figures 23, 24 and 25 where we can observe that saliency maps of naturally trained models are visually sharper and coherent because of averaging. However, using robust models increases the sparsity and produces more comprehensible saliency maps.

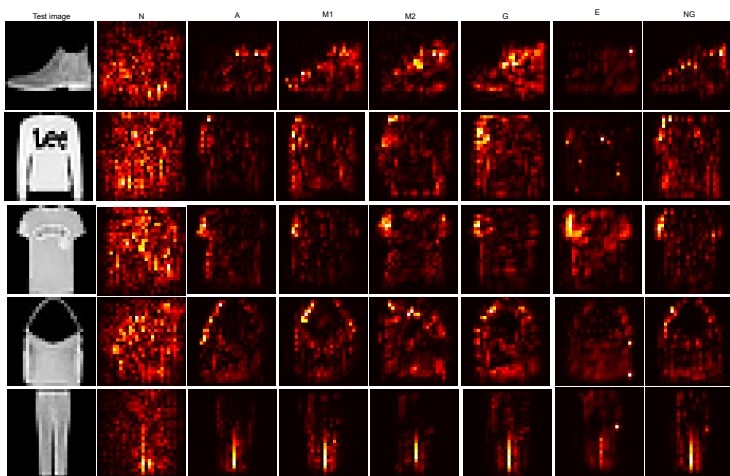

Figure 17: Additional visualization for VG (FMNIST) (N: naturally-trained, A: adversarially-trained, M1: adversarially-trained with mean-filter, M2: adversarially-trained with median-filter, G: adversarially-trained with Gaussian-filter, E: adversarially-trained with embedded filter, NG: adversarially-trained with non-local gaussian)

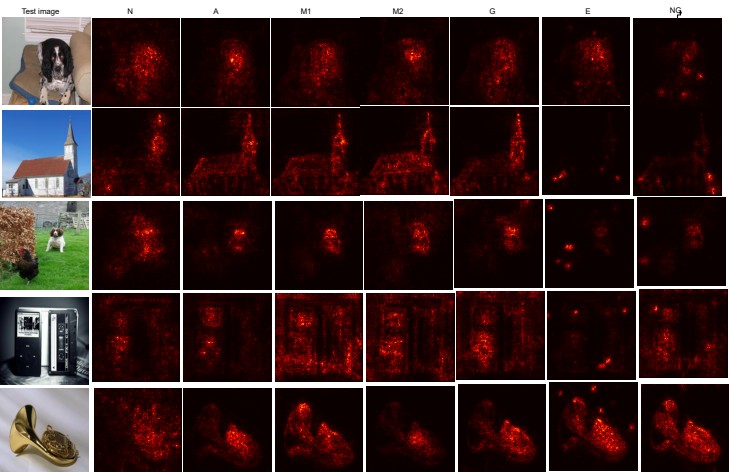

Figure 18: Additional visualization for VG (ImageNette) (N: naturally-trained, A: adversarially-trained, M1: adversarially-trained with mean-filter, M2: adversarially-trained with median-filter, G: adversarially-trained with Gaussian-filter, E: adversarially-trained with embedded filter, NG: adversarially-trained with non-local gaussian)

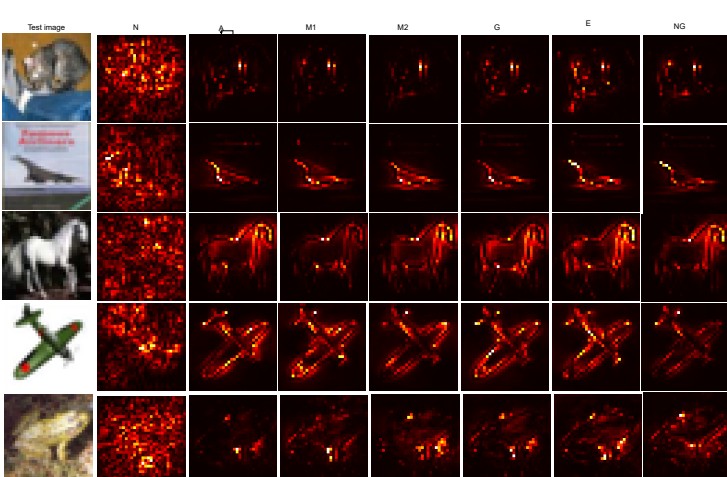

Figure 19: Additional visualization for VG (CIFAR-10) (N: naturally-trained, A: adversarially-trained, M1: adversarially-trained with mean-filter, M2: adversarially-trained with median-filter, G: adversarially-trained with Gaussian-filter, E: adversarially-trained with embedded filter, NG: adversarially-trained with non-local gaussian)

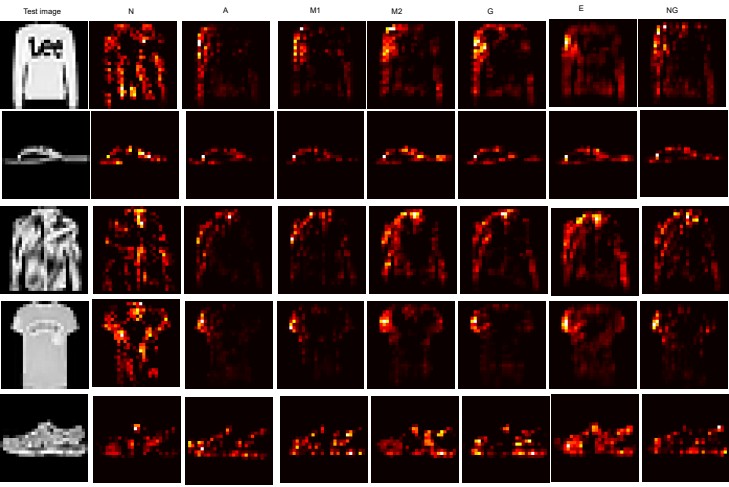

Figure 20: Saliency maps visualization on FMNIST using IG across different models (N: naturally-trained, A: adversarially-trained, M1: adversarially-trained with mean-filter, M2: adversarially-trained with median-filter, G: adversarially-trained with Gaussian-filter, E: adversarially-trained with embedded filter, NG: adversarially-trained with non-local gaussian).

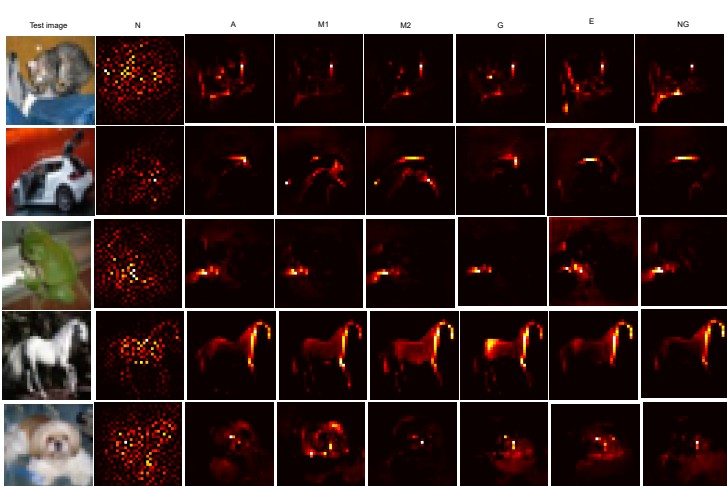

Figure 21: Saliency maps visualization on CIFAR-10 using IG across different models (N: naturally-trained, A: adversarially-trained, M1: adversarially-trained with mean-filter, M2: adversarially-trained with median-filter, G: adversarially-trained with Gaussian-filter, E: adversarially-trained with embedded filter, NG: adversarially-trained with non-local gaussian).

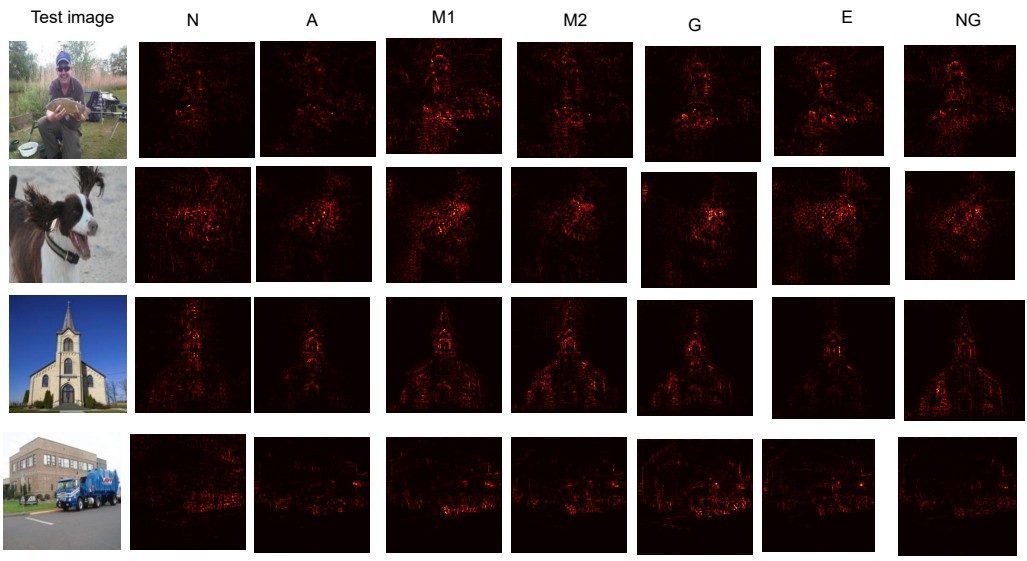

Figure 22: Saliency maps visualization on ImageNette using IG across different models (N: naturally-trained, A: adversarially-trained, M1: adversarially-trained with mean-filter, M2: adversarially-trained with median-filter, G: adversarially-trained with Gaussian-filter, E: adversarially-trained with embedded filter, NG: adversarially-trained with non-local gaussian).

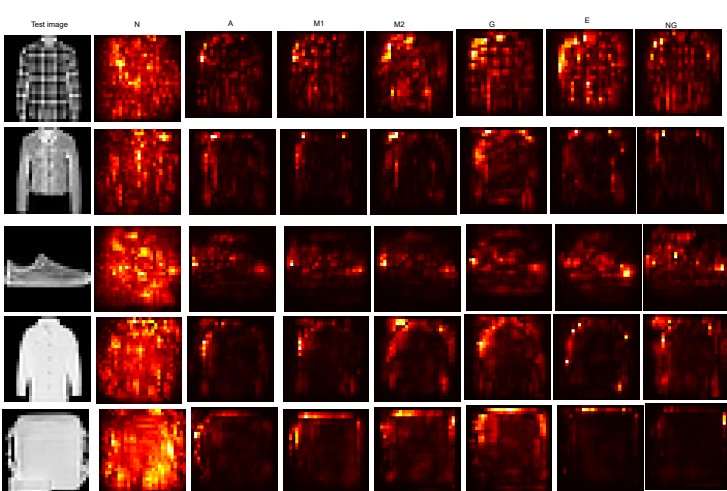

Figure 23: Saliency maps visualization on FMNIST using SmoothGrad across different models (N: naturally-trained, A: adversarially-trained, M1: adversarially-trained with mean-filter, M2: adversarially-trained with median-filter, G: adversarially-trained with Gaussian-filter, E: adversarially-trained with embedded filter, NG: adversarially-trained with non-local gaussian).

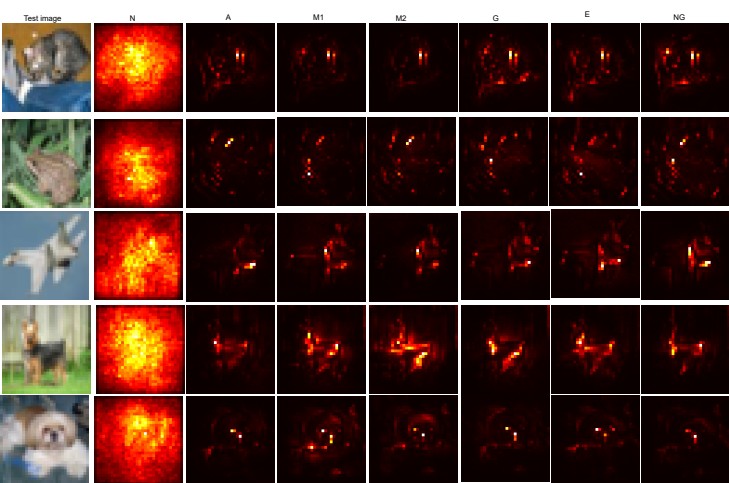

Figure 24: Saliency maps visualization on CIFAR-10 using SmoothGrad across different models (N: naturally-trained, A: adversarially-trained, M1: adversarially-trained with mean-filter, M2: adversarially-trained with median-filter, G: adversarially-trained with Gaussian-filter, E: adversarially-trained with embedded filter, NG: adversarially-trained with non-local gaussian).

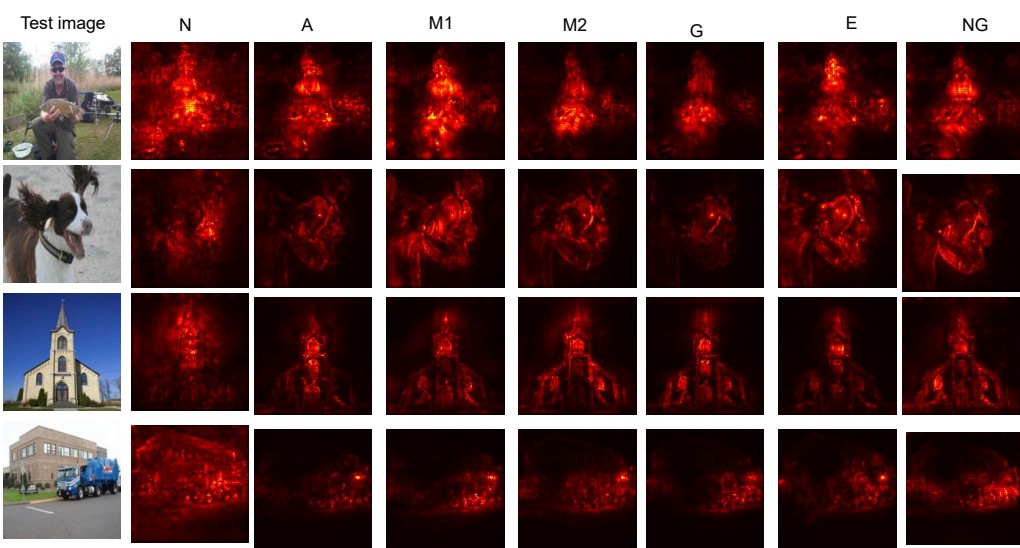

Figure 25: Saliency maps visualization on ImageNette using SmoothGrad across different models (N: naturally-trained, A: adversarially-trained, M1: adversarially-trained with mean-filter, M2: adversarially-trained with median-filter, G: adversarially-trained with Gaussian-filter, E: adversarially-trained with embedded filter, NG: adversarially-trained with non-local gaussian).

