# OpenReview forum: "Towards improving saliency map interpretability using feature map smoothing"
_ICLR.cc/2025/Conference — Submitted to ICLR 2025_

### Official Review · Reviewer_qv4W · 2024-10-31

**Soundness:** 2
**Presentation:** 3
**Contribution:** 2
**Rating:** 5
**Confidence:** 4

**Summary:**

This paper addresses challenges in the interpretability of saliency maps for image classifiers, especially those based on input gradients. Commonly used saliency map methods like Vanilla Gradient, Integrated Gradients, and SmoothGrad often fail to provide both visually clear and robust explanations. This study examines the trade-offs between stability and sparsity in saliency maps and proposes a feature map smoothing technique during adversarial training to address these issues. Experimental evaluations demonstrate that this method enhances the clarity and usability of saliency maps, making them more comprehensible and trustworthy for users. A qualitative user study supports these findings, showing that human evaluators prefer the proposed approach's explanations due to their balance between sparsity and clarity.

**Strengths:**

1. The paper presents an innovative approach by applying feature map smoothing in the context of interpretability, which has traditionally been studied mainly within adversarial robustness. By exploring its role in saliency map interpretability, the paper introduces a fresh perspective that adds significant value to explainable AI.
2. Given the indirect measures required in explainable AI due to the lack of ground truth, the experimental setup and interpretations in this paper appear extensive and credible. The design allows for robust support of the proposed method’s effectiveness in balancing clarity and stability without sacrificing sparsity.

**Weaknesses:**

1. ($\bf{Major}$) A major flaw lies in the mathematical logic presented in Section 3.1, particularly in equations (5), (6), and (7), where the claim that stability of input-gradient-based attribution ($\Delta$) is a direct result of a predictive model's sensitivity (line 171) seems overly simplified. Rather, this is mathematically false. This is because upper bound does not mean a proportional relationship. The paper lacks a discussion on whether the derived upper bounds are tight or vacuous, which would be crucial for the validity of their approach. A more nuanced analysis is needed to reinforce this section's findings.
2. ($\bf{Major}$) Sections 3.1, 3.2, and 3.3 seem loosely connected, lacking a cohesive framework. For example, if the authors’ claim regarding the direct correlation between stability and model sensitivity were accurate (I believe it's not true at this stage), then Sections 3.2 and 3.3 would need a more rigorous focus on how adversarial training and feature map smoothing concretely reduce the model's sensitivity. Presently, these sections feel more like an introduction to known techniques rather than a focused analysis supporting the main claim.
3. (Minor) The presentation of notations in Section 3 could benefit from refinement, as certain notations are repeated excessively. For example, the notation on $F(x)=H(<w,x>)$ appears too repeatedly.

**Questions:**

How confident are the authors in the tightness of the upper bound in equations (5), (6), and (7)? Are there alternative approaches to validate the proportional relationship between stability and sensitivity?

---

> ### Author Response · Authors · 2024-11-19
>
> **1. On the rigidity of the bounds of Sec 3.1**
>
> Ans: First, we want to thank the reviewer for your valuable feedback. Our intent in presenting the stability bounds was to provide a means of understanding how sensitivity (of a model) impacts the stability of saliency maps, though we recognize that the current wording may imply a more rigid or proportional relationship than intended.
>
> We agree that these bounds do not establish a strict proportional relationship between model sensitivity and attribution stability, and we did not intend for them to be interpreted as such. Rather, the bounds serve as approximate indicators, highlighting that attribution stability is influenced by model sensitivity without implying a direct proportional relationship.
>
> We have clarified this distinction in the revised paper to ensure that the bounds are seen as qualitative measures rather than strictly proportional relationships. Additionally, we have included a discussion on the tightness of these bounds in the Appendix G. Specifically, we discuss the following conditions:
> - **Model Nonlinearity and Activation Function:** Nonlinear activation functions with bounded gradients (e.g., sigmoid, tanh) tend to produce tighter bounds, while ReLU may cause looser bounds due to abrupt gradient changes near zero.
> - **Type and Magnitude of Perturbations:** Smaller perturbations, such as Gaussian noise, generally result in tighter bounds, whereas larger adversarial perturbations can cause greater output shifts, leading to looser bounds.
> - **Model Regularization:** Weight regularization smooths gradients, reducing model sensitivity and producing tighter stability bounds.
>
> All the changes are highlighted in blue in the revised paper.
>
> **2. On connection of Sec 3.1, 3.2 and 3.3**
>
> Ans: We want to thank the reviewer for the comment on the lack of cohesiveness. We have reduced the introductory tone in this section and rewritten it. The outline of these three sections are as follows:
>
> - Section 3.1 establishes the foundation for understanding how model sensitivity affects the stability of gradient-based saliency maps. Our goal here is to set up an analytical basis for using methods that can enhance stability by reducing model sensitivity.
> - Based on finding from Section 3.1, Section 3.2 introduces adversarial training as a mechanism to reduce model sensitivity by encouraging robustness to input perturbations. However, interestingly, this technique leads to sparse representations, and as a result of which, input-gradient-based explanations become more sensitive to small changes, leading to potentially unstable explanations.
> - Section 3.3 introduces feature map smoothing as the mitigation for adverse impact of adversarial training on saliency map quality. This approach also aligns with the bounds on stability derived in Section 3.1, as smoother activations lead to reduced \( \|F(\mathbf{x}') - F(\mathbf{x})\| \) and tighter stability bounds for VG, IG, and SG.
>
> 	All the changes are highlighted in blue in the revised paper.
>
> **3. How confident are the authors in the tightness of the upper bound in equations (5), (6), and (7)?**
>
> Ans: We admit our mistake in assigning proportional relationship in the bounds and have made changes to the revised paper. Our current bounds, however, are derived as approximate indicators of the relationship between stability and model sensitivity in input-gradient-based saliency maps. While they demonstrate the influence of model sensitivity on stability, these bounds are not intended as strict or tight bounds in a mathematical sense. Rather, they provide a general measure to help quantify the model-sensitivity-driven variance in the explanations produced by saliency maps. Given the conditions we have analyzed in detail Appendix G, we are confident that these bounds reasonably capture the impact of sensitivity, although they do not establish a proportional relationship.

---

> ### Comment · Reviewer_qv4W · 2024-12-03
>
> I appreciate the authors' thoughtful revisions and detailed response to my initial concerns. The improvements in the tone and presentation, particularly in Section 3.1, have significantly enhanced the clarity of the paper. By adjusting the content and expressions that previously risked misleading readers, the authors have made the arguments more precise and accessible.
>
> The clarification regarding the stability bounds and the relationship between model sensitivity and attribution stability is particularly valuable. Recognizing that the bounds serve as approximate indicators rather than implying a strict proportional relationship addresses the primary concern about the mathematical logic in Section 3.1. The inclusion of a discussion on the tightness of these bounds in the appendix adds depth to the theoretical foundation and demonstrates a commitment to further analysis. The restructured Sections 3.1, 3.2, and 3.3 now present a more cohesive narrative that effectively links model sensitivity, adversarial training, and feature map smoothing. This reorganization strengthens the logical flow of the paper and better supports the main thesis.
>
> Despite these substantial improvements, I still find that Section 3.1 lacks some of the core mathematical reasoning necessary to fully substantiate the implications for the problem at hand. While the bounds offer valuable insights, a more robust theoretical validation could further solidify the claims regarding the influence of model sensitivity on the stability of saliency maps. Nevertheless, the paper makes a meaningful contribution to the field of model interpretability. The authors provide empirical observations and reasoning that bridge seemingly unrelated aspects of explainable AI. This fresh perspective is valuable and has the potential to inspire further research and discussion within the community.
>
> In light of the authors' revisions and the paper's contributions, I have adjusted my assessment to reflect the improved quality and significance of the work.

---

### Official Review · Reviewer_24ZF · 2024-11-01

**Soundness:** 3
**Presentation:** 2
**Contribution:** 2
**Rating:** 5
**Confidence:** 3

**Summary:**

The paper proposes a method to improve the quality of saliency maps by adding a smoothing layer during adversarial training. The paper discusses that existing smoothing techniques can lead to noisy, overly sparse and incomplete model understanding. Proposed methods strike a balance between sparsity and coherency and focusing on the key regions. The paper discusses that explanation sensitivity stems from the model itself and adversarially-trained models address that sensitivity. The paper shows that their approach achieves better performance in terms of stability and faithfulness on the FMNIST, CIFAR-10 and ImageNet datasets.

**Strengths:**

+ Overall the paper the methods and introduction sections of the paper are well-written and easy to follow the story of the paper.
+ The paper evaluates their approach against a wide variety of quality metrics and methods.

**Weaknesses:**

+ In sections 3.2 - 3.3 the paper discusses that adversarial training leads to less input sensitivity and less stability. At this point in the paper it is unclear what stability is. If the model is less sensitive then the explanation will be more stable. Why is stability an issue here ? Stability and sensitivity terms don’t seem to be defined in advance. I recommend clarifying those terms before using those terms, otherwise it is hard to follow a logical chain of thoughts.
+ It is a bit hard to follow different methods compares against each other in the paper:
NG models (non-local gaussian) - what does it really mean ? Figures 4 - 9 have plots for CIFAR, FMNIST and ImageNet but ImageNet is not a dataset and the former 2 are. It is unclear what are the proposed methods and what are other methods that we are comparing against. Are M1 and M2 the methods that the authors propose and the rest are what we are comparing against ? The naming and notations of the methods makes it difficult to understand what is going on with the experimental results.
+ NG models (non-local gaussian) sounds like a model and at the same time it is evaluated in combination with ImageNet. I really recommend to have more descriptive titles in the figures. Reading the titles of the figures doesn’t tell much about how and what to interpret on the figures. At the same time the figures are full of non-intuitive acronyms and abbreviations.
Figure 4c: Why are the accuracy trajectories significantly different for M1 and M2 ? At the same time the trajectories almost overlap for  Figure 4b. Why is that the case
+ The paper discusses the limitations of the proposed approach however it is unclear why exactly authors chose 1x1 convolution after the smoothing operation.

**Questions:**

+ It is unclear what layers the smoothing should be applied. Should it be applied on the last fully connect layer ? Would the quality of explanations change if the smoothing was applied in the earlier or later layers ?
+ See weaknesses section

---

> ### Author Response · Authors · 2024-11-19
>
> **1. Definition of Stability and Sensitivity in Sections 3.2 - 3.3**
>
> Ans: To clarify: sensitivity of a model refers to the well-established definition of a model’s responsiveness to minor input changes. Stability of saliency maps, on the other hand, refers to the consistency of attributions under slight input variations. We have defined stability in Line 43 in the Introduction. While adversarial training reduces model sensitivity, it also leads to overly sparse saliency maps, which affects saliency map stability across similar inputs. Stability is thus relevant because our goal is to maintain interpretable, stable explanations. Stability and sparsity are considered to be important criteria for explanations (Chalasani et al. 2020).
>
> **2. Clarity on Dataset and Methods in Comparisons:**
>
> Ans:
>
> **Dataset Clarification:** To clarify, we use **ImageNette** in the paper and have not mentioned **ImageNet**. ImageNette is a well-established subset of ImageNet, specifically curated to represent core classes while reducing the computational burden, which aligns with our experimental goals.
>
> **Clearer Naming of Methods:** We have clarified the descriptions of each training method in Section 4.1. To clarify, the acronyms represent different model variants for three datasets: FMNIST, CIFAR-10 , and ImageNette; trained using specific training and smoothing strategies, as described in Section 4.1 (Setup). These include: **Naturally trained (N)**, **Adversarially trained (A)**, and **Adversarial training with various smoothing techniques**: **Mean-filter smoothing (M1)**, **Median-filter smoothing (M2)**, **Gaussian-filter smoothing (G)**, *Embedded-filter smoothing (E)** and **Non-local Gaussian smoothing (NG)** Because each symbol represents a different training strategy, we reuse the symbols for concise representation and saving space.
>
> **Comparison:** In this work, the goal is to evaluate the quality of saliency maps in different training strategies. Each symbol as discussed in Section 4.1 Setup introduces the training strategy and we measure quality of saliency maps for each. This is discussed in detail in Section 4.1.
>
>
> **3. Figure 4c: Why are the accuracy trajectories significantly different for M1 and M2 ? At the same time the trajectories almost overlap for Figure 4b. Why is that the case?**
>
> Ans: The variations between M1 and M2 in Figure 4c and 4b highlight how smoothing filters enhance faithfulness differently across datasets. For ImageNette (Figure 4c), the mean filter (M1) smooths the features, improving overall faithfulness by removing redundant features, while the median filter (M2) preserves more detailed features. This leads to divergent accuracy trajectories as key features are progressively removed. In CIFAR-10 (Figure 4b), where features are simpler, both filters (in addition to other smoothing filters) yield similar effects, resulting in overlapping trajectories. Across both datasets, these results however reinforce that applying smoothing filters in adversarial training enhances faithfulness of explanations compared to naturally trained models. We have added this clarification to the paper. Highlighted in blue in the revised paper in 4.2.2.
>
> **4. Why exactly did authors choose 1x1 convolution after the smoothing operation?**
>
> Ans: This design choice is inspired by the feature denoising block architecture, which integrates a smoothing operation, such as mean filter, or median filter, with a 1x1 convolution and a residual connection (See Xie et al.2019). The 1x1 convolution serves primarily as a feature combination layer. After smoothing, it linearly combines the denoised features from different channels. The added residual connection then allows the block to retain the original input features, with the 1x1 convolution adjusting how much of the smoothed output is combined with the original input. This combination improves the stability and interpretability of the saliency maps, as the model learns an end-to-end balance between noise reduction and signal retention.

---

> ### Author Response · Authors · 2024-11-19
>
> **5. It is unclear what layers the smoothing should be applied to. Should it be applied on the last fully connected layer ? Would the quality of explanations change if the smoothing was applied in the earlier or later layers?**
>
> Ans:  Thank you for your question on where smoothing filters should be applied in the network. We have discussed an ablation study in Appendix E to evaluate how the placement of smoothing filters affects the stability and sparsity of saliency maps, briefly discussed in Section 4.1 Setup.
>
> Our results show that adding the smoothing block after the **first convolutional or residual block** yields optimal balance in stability and sparsity. When smoothing filters were applied in later layers (e.g., after the second or third residual block), we observed slightly improved sparsity but a reduction in stability. Applying smoothing in earlier layers ensures that denoising occurs closer to the input, which helps stabilize feature representations across subsequent layers but with less gain in sparsity.
>
> Adding the smoothing layer at the final fully connected layer would be less effective, as this layer contains high-level, condensed representations, where denoising would not have the same stabilizing effect on intermediate features critical for interpretability. Based on these findings, we position the smoothing block after the first conv or residual block to optimize the trade-off between explanation sparsity and stability.

---

> ### Comment · Reviewer_24ZF · 2024-11-25
> **Response to authors**
>
> Thank you for the detailed response:
> In terms of:
> 1. I see papers suggesting that adversarial training leads to sparse and more stable explanations. E.g. this paper: https://proceedings.mlr.press/v119/chalasani20a/chalasani20a.pdf
> Doesn't it contradict with your answer ?
>
> Thank you for the clarifications in 2,3, 4 and 5

---

> > ### Author Response · Authors · 2024-11-25
> >
> > Thank you for the question. Chalasani et al. (2020) show that theoretically, training a 1-layer network by encouraging stability of explanations is  equivalent to adversarial training, however, they do not present any results on stability of attribution maps in multi-layer networks. Instead their work is more focused on theoretically showing why adversarial training leads to sparse saliency maps in 1-layer networks and empirically demonstrating the sparsity of attributions in multi-layer networks. We have mentioned this work in lines 105-107. In this work, we find that there is a stability-sparsity tradeoff with adversarial training: adversarial training in DNNs leads to sparse attribution maps but at the cost of stability. We discuss this tradeoff and propose the use of a smoothing layer during adversarial training to improve the stability and faithfulness of attribution maps without sacrificing sparsity.

---

> > > ### Comment · Reviewer_24ZF · 2024-11-30
> > > **Response to the authors**
> > >
> > > I think that there shouldn't be contradicting evidences between theoretical and practical analysis. Chalasani et al mentions: ` ... adversarial training are both concise (due to the sparseness of the models), and stable.` If your findings contradict other theoretical work then you might want to discuss the theory behind your work. This way you can compare the theory of stability in your work vs in Chalasani et. al.

---

> > > > ### Author Response · Authors · 2024-11-30
> > > >
> > > > Thank you for your question. Regarding Chalasani “adversarial training are both concise (due to the sparseness of the models), and stable”:  they mention that this holds true only for single-layer networks and not in multilayer networks. In DNNs, only conciseness (sparseness) is achieved. This has been demonstrated in similar other works (Etmann et al., 2019; Zhang & Zhu, 2019). In Section 4.2.1, we follow the exact setup of Chalasani et al. and demonstrate that with adversarial training, there is an increase in sparsity but this comes at the expense of stability, suggesting an inverse relationship between the sparsity and stability of saliency maps. With our proposed approach of using feature-map smoothing with adversarial training, we can obtain concise and stable saliency maps. This is supported by our theoretical analysis, as we discuss in Section 3.3 that the smoothing operation yields tighter bound of stability norm ∥F (x′) − F (x)∥, providing stable explanations for VG, IG, and SG.

---

### Official Review · Reviewer_AESz · 2024-11-02

**Soundness:** 2
**Presentation:** 2
**Contribution:** 2
**Rating:** 5
**Confidence:** 4

**Summary:**

The paper explores enhancing the interpretability of saliency maps for image classification networks by balancing sparsity and stability. Traditional gradient-based attribution methods, like Vanilla Gradient and SmoothGrad, aim to provide clear explanations but often face issues with either excessive sparsity or instability, particularly under adversarial training. To address this, this paper proposes adding a smoothing layer during adversarial training to maintain sparse yet stable saliency maps. Their experiments show that this approach enhances the stability and faithfulness of explanations without compromising on sparsity. A user study confirms that human evaluators prefer the explanations generated by this method, as they find them clearer and less noisy than those from natural and adversarially trained models.

**Strengths:**

This paper presents two strengths.
1. It introduces an interesting approach that effectively balances stability and sparsity in saliency maps by applying a smoothing layer during adversarial training.
2. This paper includes a comprehensive user study that validates the method's effectiveness from a human perspective. The study shows that users find these explanations more interpretable and trustworthy.

**Weaknesses:**

The following questions hinder the contribution of this paper.
1. Though applying feature map smoothing and adversarial training improves the sparsity and stability of the model explanations, it is unclear whether this training technique hinders the accuracy. Besides, this XAI method requires retraining the model, which limits the generalization ability when the models can not be trained.
2. The proposed smoothing technique is simply adapted from the Feature-map Smoothing paper (Xie et al., 2019), hindering the novelty.
3. There are many typos in this paper, and the author is encouraged to go through the paper carefully. For instance, in Line 261, page 5, ImageNette -> ImageNet. Besides, the author misused  /cite, /citep and /citet, see Line 414 in Page 8 for an example.

**Questions:**

Please refer to the Weaknesses section.

---

> ### Author Response · Authors · 2024-11-19
>
> **1. On Impact of Smoothing Filters on Accuracy**
>
> Ans: Thank you for raising this question. In Appendix D, we examine the effects of applying smoothing filters on both benign (natural) and robust (adversarial) accuracy across FMNIST, CIFAR-10, and ImageNette. Our results show that applying smoothing filters alone does not significantly alter natural or robust accuracy (approximately ( \pm 3\% ) change) across datasets, suggesting that these filters primarily serve to stabilize feature maps without drastically affecting model performance boundaries. When combining smoothing filters with adversarial training, we observed improvements in robust accuracy, especially on FMNIST and CIFAR-10. However, on ImageNette, which contains more complex features, there was a notable drop in benign accuracy when smoothing filters were applied during adversarial training, likely due to the model prioritizing robustness over fine-grained accuracy.
>
> **2. On-"this XAI method requires retraining the model, which limits the generalization ability when the models can not be trained."**
>
> Ans: Thank you for this comment. Our work does not propose a new XAI method; rather, it analyzes why the quality of explanations is often poor in naturally trained models and explores improvements that can be achieved through smoothing and adversarial training. Many existing XAI methods have shown limitations, often failing crucial sanity checks, which calls into question their faithfulness, and reliability. As noted by Ilyas et al. 2019, explanations that are both meaningful and faithful to the model's decision-making cannot be obtained in isolation from the model's training. This principle is central to our approach, as we aim to enhance explanation quality by refining the training method. By integrating smoothing and adversarial training, we produce explanations that are more faithful and comprehensible to end-users for existing methods, rather than creating a new post hoc method.
>
> **3. The proposed smoothing technique is simply adapted from the Feature-map Smoothing paper (Xie et al., 2019), hindering the novelty.**
>
> Ans: Thank you for this observation. We acknowledge that our smoothing technique is based on the approach proposed by Xie et al. (2019). Our goal, however, is not to introduce a new smoothing method, but to conduct a detailed analysis of how applying such feature-map smoothing during adversarial training can enhance the interpretability of post-hoc explanations, specifically in terms of sparsity, stability, and faithfulness. Prior works have shown that adversarial training produces sparser maps (albeit with a different kind of study), however, we observe that only using adversarial training affects explanation stability and through extensive experimentation, we evaluate how different smoothing filters affect the quality of explanations. This work hence focuses on the interpretability aspect, and combined with our empirical findings across multiple datasets and metrics, provides new insights into enhancing faithfulness of explanations.
>
> **4. Typos in the paper**
>
> Ans: To clarify, we use **ImageNette** in the paper and have not mentioned **ImageNet**. ImageNette is a well-established subset of ImageNet, specifically curated to represent core classes while reducing the computational burden, which aligns with our experimental goals. We have fixed the typos in using citation as suggested by the reviewer.

---

> > ### Comment · Reviewer_AESz · 2024-11-26
> > **Response to authors.**
> >
> > I appreciate the response by the authors, which partially addresses some concerns. However, the main issue of lacking novelty and generalization ability remains unresolved. Thus, I maintain my rating and encourage the authors to refactor the paper for greater clarity and contribution.

---

> > > ### Author Response · Authors · 2024-11-26
> > >
> > > Thank you for your follow-up comments. Below, we clarify our contributions further and address the concerns around novelty and generalization ability:
> > >
> > > **1. On the Question of Novelty:** We understand your concern about novelty, particularly since our smoothing technique draws from Xie et al. (2019). However, our work does not aim to innovate on smoothing techniques per se but rather **uniquely integrates feature-map smoothing with adversarial training** to address long-standing issues in saliency maps, such as sparsity, stability, and faithfulness. We systematically analyzed the stability-sparsity tradeoff in adversarial training and demonstrated improvements in saliency map quality by integrating feature-map smoothing. These analysis and insights on the **trade-off between stability and sparsity in saliency maps** have not been undertaken in prior works. We believe these contributions provide novel insights into improving the quality of saliency maps through training interventions.
> > >
> > >
> > > **2. On Generalization Ability:** We agree that requiring retraining the model can limit applicability. However, our work targets scenarios where retraining is feasible. As noted in our response and supported by works like Ilyas et al. (2019), explanations that align with a model's true decision-making process often require training-level interventions. Our approach demonstrates that when adversarial training is desirable, feature map smoothing can be effectively integrated to enhance the quality of saliency maps. The results could also guide the design of models optimized for explainability.
> > >
> > > If these points are unclear in the manuscript, we welcome your additional guidance for improving it and hope you will re-examine and reconsider your score.
> > >
> > > Thank you again for your feedback.

---

### Official Review · Reviewer_EXUX · 2024-11-03

**Soundness:** 3
**Presentation:** 3
**Contribution:** 2
**Rating:** 6
**Confidence:** 4

**Summary:**

The authors propose a solution to improve saliency maps by smoothening the feature maps during training.
The solution aims to address instability of the maps incurred by adversarial training.
The solution is tested through experiments with ResNet-18 on three datasets (CIFAR10, FMNIT, ImageNette).

**Strengths:**

- The authors offers new insights about the trade-off between stability and sparsity in saliency maps.
- The solution provided seems helpful to improve the stability (and potentially the faithfulness), with minimal impact on the sparsity.

**Weaknesses:**

- The contribution is limited in terms of depth and impact.
The method proposed requires adaptation to model training and architecture and hence cannot be used as a generic attribution method.
- The analysis is limited in terms of architectures tested.
The method only supports computer-vision models. Moreover, there were no tests with ViT, now a major architecture in CV.
- There was no in-depth analysis about why smoothening is required besides showing some experimental results.
My concern is that smoothening incurs significant change to the architecture by adding new layers. Besides an explicit 1x1 convolution added, the smoothening filter is implemented as an implicit KxK conv layer. This makes most comparisons unfair. Even if the parameters of the smoothenng kernel are frozen, however, the kernel itself expands the receptive field. By taking into account all layers with strides, the cumulative expansion of the RF becomes very significant. The perceived advantages might simply be a result of this expansion. The authors did not control for such effects.

There were frequent writing issues with respect to articles and citation format. For ex:
we demonstrate this smoothing => that
within same object => the same
for the designing models => remove “the”
Please use \citep when the citation is not a natural part of the text and should be inside parentheses, and use \citet when the author names are part of the sentence with sound grammar.

**Questions:**

How would the proposed method work for ViT models? And how could smoothening be implemented in NLP transformers?

---

> ### Author Response · Authors · 2024-11-19
>
> **1. On contribution-”proposed method requiring adaptation to model training and cannot be used as a generic attribution method”.**
>
> Ans: Thank you for your feedback regarding the depth and adaptability of our approach. Our primary contribution is not aimed at proposing a new generic attribution method but rather at analyzing and enhancing the interpretability of explanations within robust, adversarially trained models. We investigate why explanations in naturally trained models lack quality in terms of sparsity, stability and faithfulness and demonstrate how feature-map smoothing, when integrated with adversarial training, improves such explanation quality.
>
> As highlighted in Ilyas et al. 2019, explanations that are both meaningful and faithful to the model's decision-making cannot be developed in isolation from the model's training process. This principle is central to our approach, as we aim to enhance explanation quality by refining the training method. In line with this, our approach shows that for the same explanation methods (VG, IG, SG), a robust model with feature map smoothing yields more reliable and comprehensible explanations than with a naturally trained model. This is not a one-size-fits-all solution but demonstration that adapting a model training is essential for creating more meaningful and reliable explanations.
>
> **2. On support for computer-vision models only and questions on ViT.**
>
> Ans: Thank you for your comment. Our analysis primarily focused on CNNs, as they are still widely used in computer vision research and provide a well-established baseline for evaluating the impact of smoothing and adversarial training on explanation quality.
>
> We acknowledge that ViTs have become a significant architecture in computer vision. Due to the distinct structural differences between CNNs and ViTs, such as the presence of self-attention layers in ViTs, adapting and optimizing our smoothing technique specifically for ViTs requires additional research and experimentation. This is an area we are actively exploring and plan to address in future work to extend the applicability of our method.
>
> However, we believe that our findings on CNNs serve as a foundational step in understanding how the quality of explanations is tied to the model training strategy and the role of feature-map smoothing in enhancing explanation quality in robust models. Testing on more architectures, including ViTs, would further strengthen this work, and we appreciate your suggestion as a direction for future research.

---

> ### Author Response · Authors · 2024-11-19
>
> **3. On Receptive Field Expansion**
>
> Ans: Thank you for this observation regarding the effect of smoothing filters on receptive field expansion. While certain smoothing filters (e.g. mean, Gaussian) operate over local neighborhoods, they are non-learnable and are not equivalent to traditional K×K convolutional layers with learnable parameters. Instead, they serve to reduce noise in the feature maps without significantly increasing model capacity or altering representational power. For non-local filters (e,g, Non-local Gaussian and Embedded Gaussian), the smoothing effect is based on feature similarity across the entire feature map, rather than a localized convolution. This approach captures long-range dependencies but might not expand the receptive field in the same spatial manner as a series of convolutional layers with strides.
>
> To measure the receptive field effects, we conducted an additional experiment on CIFAR-10 where we modified the feature smoothing block so that it performs only a convolution (identify or randomly initialized). This modified setup ensures that there is only an expansion of the receptive field without filtering operations and it can provide a baseline study to analyze the effect of receptive field expansion on its own.
>
> Following are the results for Vanilla Gradient (VG) when compared with the best performing model, where M2: adversarial training with median smoothing, Identity: adversarial training with feature smoothing block consisting of identity convolution but no smoothing filter and Random: adversarial training with feature smoothing block consisting of randomly initialized convolution but no smoothing filters,
>
> | **Models**                                     | **M2** | **Identity** | **Random** |
> |------------------------------------------------|--------|--------------|------------|
> | **Sparsity (dG) (higher is better)**           | **0.18**   | 0.16         | 0.15       |
> | **Relative input stability (dRIS) (lower is better)** | **-0.68**  | -0.41        | -0.36      |
> | **Relative output stability (dROS) (lower is better)** | 0.21   | 0.07         | **0.06**       |
> | **Relative representation stability (dRRS) (lower is better)** | 0.43   | **0.41**         | 0.43       |
>
>
> The results in the table show that:
>
> - The `M2` model still achieves the best sparsity, indicating that the smoothing operation in addition to the convolutional operation helps the model to learn a smaller number of discriminative features.
>
> - The `M2` model performs significantly better in input stability. This indicates that smoothing filters provide stability in saliency maps with respect to input.
> - Interestingly, the `M2` model does not achieve the best score in output stability. This suggests that while smoothing helps in stabilizing attributions with respect to inputs and internal representations, it might not directly translate to stability at the model's output layer. The expanded receptive field introduced by the identity or random convolutions likely contributes to this improvement.
> - The `Identity` model achieves the best representation stability but only marginally outperforming `M2`.
>
> Overall, the inclusion of smoothing operations still provides a competitive advantage in improving the quality of saliency maps with respect to sparsity, input stability and representation stability. We have included this study in the revised paper Appendix H.
>
> **4. On citation typos**
>
> Ans: We want to thank the reviewer for the suggestion. We have fixed the citation typos.

---

> > ### Comment · Reviewer_EXUX · 2024-11-23
> >
> > I appreciate the response by the authors which to some extend addresses my concerns.
> > I am increasing my review score accordingly.

---

### Meta-Review · Area_Chair_kZaG · 2024-12-21

**Metareview:**

The reviewers raised several weaknesses: There is insufficient justification for the smoothing technique, and the addition of new layers, especially the smoothing filter, unfairly alters the architecture. Furthermore, the paper's terminology around stability and sensitivity is unclear, and the experimental results are difficult to follow due to poor figure titles and unclear method comparisons. The mathematical logic in Sections 3.1–3.3 is overly simplistic, with unsubstantiated claims about the relationship between stability and model sensitivity. Additionally, the paper suffers from numerous writing issues, such as typos, citation formatting errors, and unclear notations, which affect overall clarity. The proposed method is also not sufficiently novel, as it closely resembles existing work, and its impact on model accuracy and generalization is not fully addressed.

**Additional Comments On Reviewer Discussion:**

Authors and reviewers discussed. Some of the concerns were addressed so two reviewers raised their score. However, some reviewers still have concerns after the discussion. For instance Reviewer qv4W says: "I still find that Section 3.1 lacks some of the core mathematical reasoning necessary to fully substantiate the implications for the problem at hand."

In the end, the ratings are 6,5,5,5. Since no reviewer is very positive about the work, it will not be accepted.

---

### Decision · Program_Chairs · 2025-01-22

Reject